# Learn to Categorize or Categorize to Learn?
# Self-Coding for Generalized Category Discovery

**Sarah Rastegar, Hazel Doughty,**\* **Cees G. M. Snoek**
University of Amsterdam

## Abstract

In the quest for unveiling novel categories at test time, we confront the inherent limitations of traditional supervised recognition models that are restricted by a pre-defined category set. While strides have been made in the realms of self-supervised and open-world learning towards test-time category discovery, a crucial yet often overlooked question persists: what exactly delineates a *category*? In this paper, we conceptualize a *category* through the lens of optimization, viewing it as an optimal solution to a well-defined problem. Harnessing this unique conceptualization, we propose a novel, efficient and self-supervised method capable of discovering previously unknown categories at test time. A salient feature of our approach is the assignment of minimum length category codes to individual data instances, which encapsulates the implicit category hierarchy prevalent in real-world datasets. This mechanism affords us enhanced control over category granularity, thereby equipping our model to handle fine-grained categories adeptly. Experimental evaluations, bolstered by state-of-the-art benchmark comparisons, testify to the efficacy of our solution in managing unknown categories at test time. Furthermore, we fortify our proposition with a theoretical foundation, providing proof of its optimality. Our code is available at: `https://github.com/SarahRastegar/InfoSieve`.

## 1 Introduction

The human brain intuitively classifies objects into distinct categories, a process so intrinsic that its complexity is often overlooked. However, translating this seemingly innate understanding of categorization into the realm of machine learning opens a veritable Pandora's box of differing interpretations for *what constitutes a category?* [1, 2]. Prior to training machine learning models for categorization tasks, it is indispensable to first demystify this concept of a *category*.

In the realm of conventional supervised learning [3–7], each category is represented by arbitrary codes, with the expectation that machines produce corresponding codes upon encountering objects from the same category. Despite its widespread use, this approach harbors several pitfalls: it suffers from label inconsistency, overlooks category hierarchies, and, as the main topic of this paper, struggles with open-world recognition.

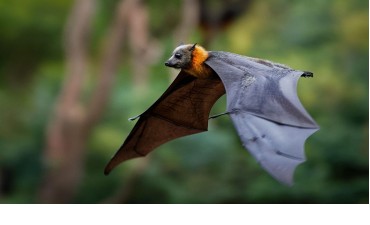

Figure 1: **What is the correct category?** Photo of a *flying fox fruit bat*. This image can be categorized as *bat*, *bird*, *mammal*, *flying bat*, and other categories. How should we define which answer is correct? This paper uses self-supervision to learn an implicit category code tree that reveals different levels of granularity in the data.

**Pitfall I: Label Inconsistency.** Assessing a model's performance becomes problematic when category assignments are subject to noise [8, 9] or exhibit arbitrary variation across different datasets. For example, if a zoologist identifies the bird in Figure 1 as a *flying fox fruit bat*, it is not due to a misunderstanding of what constitutes a *bird* or a *dog*. Rather, it signifies a more nuanced understanding of these categories. However, conventional machine learning

---

\*Currently at Leiden University

37th Conference on Neural Information Processing Systems (NeurIPS 2023).

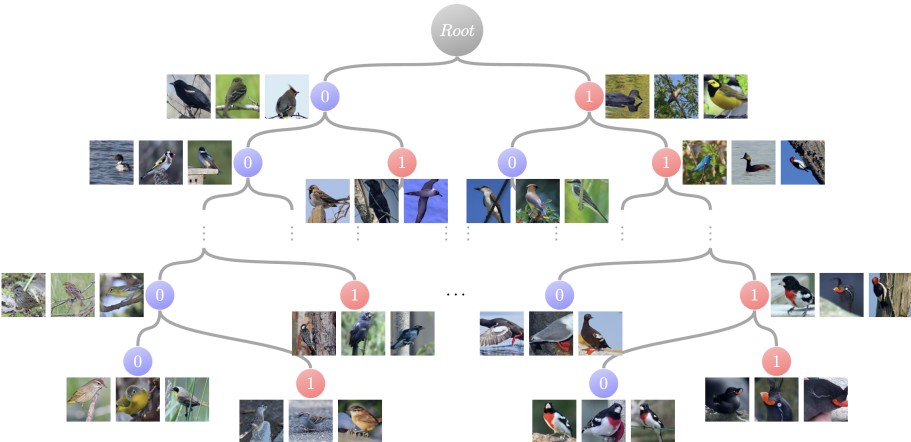

Figure 2: **The implicit binary tree our model finds to address samples.** Each leaf in the tree indicates a specific sample, and each node indicates the set of its descendants' samples. For instance, the node associated with '11...11' is the set of all birds with red beaks, while its parent is the set of all birds with red parts in their upper body.

models may penalize such refined categorizations if they deviate from the pre-defined ground-truth labels. This work addresses this limitation by assigning category codes to individual samples. These codes not only prevent over-dependence on specific labels but also facilitate encoding similarities across distinct categories, paving the way for a more robust and nuanced categorization.

**Pitfall II: Category Hierarchies.** A prevalent encoding method, such as one-hot target vectors, falls short when addressing category hierarchies. While we, as humans, intuitively distinguish the categories of *plane* and *dog* as more disparate than *cat* and *dog*, our representations within the model fail to convey this nuanced difference, effectively disregarding the shared semantics between the categories of *cat* and *dog*. While some explorations into a category hierarchy for image classification have been undertaken [10–14], these studies hinge on an externally imposed hierarchy, thus limiting their adaptability and universality. This paper proposes a self-supervised approach that enables the model to impose these implicit hierarchies in the form of binary trees into their learned representation. For instance, Figure 2 shows that we can address each sample in the dataset with its path from the root of this implicit tree, hence associating a category code to it. We show theoretically that under a set of conditions, all samples in a category have a common prefix, which delineates category membership.

**Pitfall III: Open World Recognition.** The final problem we consider is the encounter with the open world [15–17]. When a model is exposed to a novel category, the vague definition of *category* makes it hard to deduce what will be an unseen new category. While open-set recognition models [18–22] can still evade this dilemma by rejecting new categories, Novel Class Discovery [23–26] or Generalized Category Discovery [27–31] can not ignore the fundamental flaw of a lack of definition for a *category*. This problem is heightened when categories are fine-grained [32, 33] or follow a long-tailed distribution [34–37].

In this paper, we confront these challenges by reframing the concept of a *category* as the solution to an optimization problem. We argue that categories serve to describe input data and that there is not a singular *correct* category but a sequence of descriptions that span different levels of abstraction. We demonstrate that considering categorization as a search for a sequence of category codes not only provides more flexibility when dealing with novel categories but also, by leveraging sequences, allows us to modulate the granularity of categorization, proving especially beneficial for fine-grained novel categories. Subsequently, we illustrate how to construct a framework capable of efficiently approximating this solution. Our key contributions are as follows:

- *Theoretical.* We conceptualize a *category* as a solution to an optimization problem. We then demonstrate how to fine-tune this optimization framework such that its mathematical solutions align with the human-accepted notion of *categories*. Furthermore, under a set of well-defined constraints, we establish that our method theoretically yields an optimal solution.
- *Methodological.* Based on the theory we developed, we propose a practical method for tackling the generalized category discovery problem, which is also robust to different category granularities.
- *Experimental.* We empirically show that our method outperforms state-of-the-art generalized category discovery and adapted novel class discovery methods on fine-grained datasets while performing consistently well on coarse-grained datasets.

Before detailing our contributions, we first provide some background on category discovery to better contextualize our work.

## 2 Background

The *Generalized Category Discovery* problem introduced by Vaze et al. [27] tries to categorize a set of images during inference, which can be from the known categories seen during training or novel categories. Formally, we only have access to $\mathcal{Y}_S$ or seen categories during training time, while we aim to categorize samples from novel categories or $\mathcal{Y}_U$ during test time. For the Novel Class Discovery problem, it is assumed that $\mathcal{Y}_S \cap \mathcal{Y}_U = \emptyset$. However, this assumption could be unrealistic for real-world data. So Vaze et al. [27] proposed to use instead the more realistic Generalized Category Discovery assumption in which the model can encounter both seen and unseen categories during test time. In short, for the Generalized Category Discovery problem, we have $\mathcal{Y}_S \subset \mathcal{Y}_U$.

One major conundrum with both Generalized Category Discovery and Novel Category Discovery is that the definition of the *category* has remained undetermined. This complication can be overlooked when the granularity of categories at test time is similar to training time. However, for more realistic applications where test data may have different granularity from training data or categories may follow a long-tailed distribution, the definition of *category* becomes crucial. To this end, in the next section, we formulate categories as a way to abstract information in the input data.

## 3 An Information Theory Approach to Category Coding

To convert a subjective concept as a *category* to a formal definition, we must first consider why categorization happens in the first place. There are many theories regarding this phenomenon in human [38–40] and even animal brains [41–43]. One theory is *categorization* was a survival necessity that the human brain developed to retrieve data as fast and as accurately as possible [44]. Studies have shown that there could be a trade-off between retrieval speed and accuracy of prediction in the brain [45–47]. Meanwhile, other studies have shown that the more frequent categories can be recognized in a shorter time, with more time needed to recognize fine-grained nested subcategories [48, 49]. These studies might suggest shorter required neural pulses for higher hierarchy levels. Inspired by these studies, we propose categorization as an optimization problem with analogous goals to the human brain. We hypothesize that we can do the category assignment to encode objects hierarchically to retrieve them as accurately and quickly as possible.

**Notation and Definitions.** Let us first formalize our notation and definitions. We denote the input random variable with $X$ and the category random variable with $C$. The category code random variable, which we define as the embedding sequence of input $X^i$, is denoted by $z^i = z_1^i z_2^i \cdots z_L^i$, in which superscript $i$ shows the $i$th sample, while subscript $L$ shows the digit position in the code sequence. In addition, $I(X; Z)$ indicates the mutual information between random variables $X$ and $Z$ [50, 51], which measures the amount of *information* we can obtain for one random variable by observing the other one. Since category codes are sequences, algorithmic information theory is most suitable for addressing the problem. We denote the algorithmic mutual information for sequences x and z with $I_{alg}(\mathrm{x} : \mathrm{z})$, which specifies how much information about sequence x we can obtain by observing sequence z. Both Shannon and algorithmic information-theory-based estimators are useful for hierarchical clustering [52–56], suggesting we may benefit from this quality to simulate the implicit category hierarchy. A more in-depth discussion can be found in the supplemental.

### 3.1 Maximizing the Algorithmic Mutual Information

Let's consider data space $\mathcal{D} = \{X^i, C^i : i \in \{1, \cdots N\}\}$ where $X$s are inputs and $C$s are the corresponding category labels.

**Lemma 1** *For each category c and for $X^i$ with $C^i = c$, we can find a binary decision tree $\mathcal{T}_c$ that starting from its root, reaches each $X^i$ by following the decision tree path. Based on this path, we assign code $c(X^i) = c_1^i c_2^i \cdots c_M^i$ to each $X^i$ to uniquely define and retrieve it from the tree. $M$ is the length of the binary code assigned to the sample.*

Proof of Lemma 2 is provided in the supplemental. Based on this lemma, we can find a forest with categories $c$ as the roots and samples $X^i$s as their leaves. We apply the same logic to find a super decision tree $\mathbf{T}$ that has all these category roots as its leaves. If we define the path code of category $c$ in this super tree by $p(c) = p_1^c p_2^c \cdots p_K^c$ where $K$ is the length of the path to the category $c$; we find the path to each $X^i$ in the supertree by concatenating its category path code with its code in the category decision tree. So for each input $X^i$ with category $c$ we define an address code as $q_1^i q_2^i \cdots q_{K+M}^i$ in which $q_j^i = p_j^c$ for $j \leq K$ and $q_j^i = c_{j-K}^i$ for $j > K$. Meanwhile, since all $X^i$s are the descendants

of root $c$ in the $\mathcal{T}_c$ tree, we know there is one encoding to address all samples, in which samples of the same category share a similar prefix. Now consider a model that provides the binary code $\mathrm{z}^i = z_1^i \cdots z_L^i$ for data input $X^i$ with category $c$, let's define a valid encoding in Definition 2.

**Definition 1** *A valid encoding for input space $\mathcal{X}$ and category space $\mathcal{C}$ is defined as an encoding that uniquely identifies every $X^i \in \mathcal{X}$. At the same time, for each category $c \in \mathcal{C}$, it ensures that there is a sequence that is shared among all members of this category but no member out of the category.*

As mentioned before, if the learned tree for this encoding is isomorph to the underlying tree $\mathbf{T}$, we will have the necessary conditions that Theorem 1 provides.

**Theorem 1** *For a learned binary code $\mathrm{z}^i$ to address input $X^i$, uniquely, if the decision tree of this encoding is isomorph to underlying tree $T$, we will have the following necessary conditions:*

1. $I_{alg}(\mathrm{z} : x) \geq I_{alg}(\tilde{z} : x) \quad \forall \tilde{z}, \tilde{z}$ *is a valid encoding for $x$*
2. $I_{alg}(\mathrm{z} : c) \geq I_{alg}(\tilde{z} : c) \quad \forall \tilde{z}, \tilde{z}$ *is a valid encoding for $x$*

Proof of Theorem 1 is provided in the supplemental. Optimizing for these two measures provides an encoding that satisfies the necessary conditions. However, from the halting Theorem [57], this optimization is generally not computable [58–61].

**Theorem 1 Clarification**. The first part of Theorem 1 states that if there is an implicit hierarchy tree, then for any category tree that is isomorph to this implicit tree, the algorithmic mutual information between each sample and its binary code generated by the tree will be maximal for the optimal tree. Hence, maximizing this mutual information is a necessary condition for finding the optimal tree. This is equivalent to finding a tree that generates the shortest-length binary code to address each sample uniquely. The second part of Theorem 1 states that for the optimal tree, the algorithmic mutual information between each sample category and its binary code will be maximum. Hence, again, maximizing this mutual information is a necessary condition for finding the optimal tree. This is equivalent to finding a tree that generates the shortest-length binary code to address each category uniquely. This means that since the tree should be a valid tree, the prefix to the unique address of every category sample $c$ should be the shortest-length binary code while not being the prefix of any sample from other categories.

**Shannon Mutual Information Approximation**. We can approximate these requirements using Shannon mutual information instead if we consider a specific set of criteria. First, since Shannon entropy does not consider the relationship between separate bits or $z_j^i$s, we convert each sequence to an equivalent random variable number by considering its binary digit representation. To this end, we consider $Z^i = \sum_{k=1}^m \frac{z_k^i}{2^k}$, which is a number between 0 and 1. To replace the first item of Theorem 1 by its equivalent Shannon mutual information, we must also ensure that z has the minimum length. For the moment, let's assume we know this length by the function $l(X^i) = l_i$. Hence instead of $Z^i$, we consider its truncated form $Z_{l_i}^i = \sum_{k=1}^{l_i} \frac{z_k^i}{2^k}$. This term, which we call the address loss function, is defined as follows:

$$\mathcal{L}_{\mathrm{adr}} = -\frac{1}{N} \sum_{i=0}^N I(X^i; Z_{l_i}^i) \quad s.t. \quad Z_{l_i}^i = \sum_{k=1}^{l_i} \frac{z_k^i}{2^k} \text{ and } \forall k, z_k^i \in \{0, 1\}. \tag{1}$$

We can approximate this optimization with a contrastive loss. However, there are two requirements that we must consider; First, we have to obtain the optimal code length $l_i$, and second, we have to ensure $z_k^i$ is binary. In the following sections, we illustrate how we can satisfy these requirements.

### 3.2 Category Code Length Minimization

To find the optimal code lengths $l_i$ in Equation 1, we have to minimize the total length of the latent code. We call this loss $\mathcal{L}_{\mathrm{length}}$, which we define as $\mathcal{L}_{\mathrm{length}} = \frac{1}{N} \sum_{i=0}^N l_i$. However, since the $l_i$ are in the subscripts of Equation 1, we can not use the conventional optimization tools to optimize this length. To circumvent this problem, we define a binary mask sequence $\mathrm{m}^i = m_1^i m_2^i \cdots m_L^i$ to simulate the subscript property of $l_i$. Consider a masked version of $\mathrm{z}^i = z_1^i \cdots z_L^i$, which we will denote as $\tilde{\mathrm{z}}^i = \tilde{z}_1^i \cdots \tilde{z}_L^i$, in which for $1 \leq k \leq L$, we define $\tilde{z}_k^i = z_k^i m_k^i$. The goal is to minimize the number of ones in sequence $\mathrm{m}^i$ while forcing them to be at the beginning of the sequence. One way to ensure this is to consider the sequence $\bar{\mathrm{m}}^i = (m_1^i 2^1)(m_2^i 2^2) \cdots (m_L^i 2^L)$ and minimize its $L_p$ Norm for $p \geq 1$.

This will ensure the requirements because adding one extra bit has an equivalent loss of all previous bits. In the supplemental, we provide a more rigorous explanation.

$$\mathcal{L}_{\text{length}} \approx \frac{1}{N} \sum_{i=0}^{N} \| \bar{m}^i \|_p .$$ (2)

We extract the mask from the input $X^i$, i.e., $m^i = Mask(X^i)$. Mask digits should also be binary, so we need to satisfy their binary constraints, which we will address next.

**Satisfying Binary Constraints.** Previous optimizations are constrained to two conditions, *Code Constraint:* $\forall z_k^i, z_k^i = 0$ *or* $z_k^i = 1$ and *Mask Constraint:* $\forall m_k^i, m_k^i = 0$ *or* $m_k^i = 1$. We formulate each constraint in an equivalent Lagrangian function to make sure they are satisfied. For the binary code constraint we consider $f_{\text{code}}(z_k^i) = (z_k^i)(1 - z_k^i) = 0$, which is only zero if $z_k^i = 0$ or $z_k^i = 1$. Similarly, for the binary mask constraint, we have $f_{\text{mask}}(m_k^i) = (m_k^i)(1 - m_k^i) = 0$. To ensure these constraints are satisfied, we optimize them with the Lagrangian function of the overall loss.

### 3.3 Aligning Category Codes using Supervision Signals

The second item in Theorem 1 shows the necessary condition for maximizing mutual information with a category. If we replace algorithmic mutual information with its Shannon cousin, we will have:

$$\mathcal{L}_{\text{Cat}} = -\frac{1}{N} \sum_{i=0}^{N} I(c^i; Z_{l_i}^i) \quad s.t. \quad Z_{l_i}^i = \sum_{k=1}^{l_i} \frac{z_k^i}{2^k} \text{ and } \forall k, z_k^i \in \{0, 1\}.$$ (3)

Subject to satisfying the binary constraints. Note that the optimal lengths might differ for optimizing information based on categories or input. However, here we consider the same length for both scenarios for simplicity.

**Overall Loss.** Putting all these losses and constraints together, we will reach the constrained loss:

$$\mathcal{L}_{\text{constrained}} = \mathcal{L}_{\text{adr}} + \delta\mathcal{L}_{\text{length}} + \gamma\mathcal{L}_{\text{Cat}} \quad s.t. \quad \forall k, i \ f_{\text{code}}(z_k^i) = 0, \quad \forall k, i \ f_{\text{mask}}(m_k^i) = 0.$$ (4)

Note that when we do not have the supervision signals, we can consider $\gamma = 0$ and extract categories in an unsupervised manner. In the supplemental, we have shown how to maximize this function based on the Lagrange multiplier. If we indicate $\mathcal{L}_{\text{code\_cond}} = \sum_{i=0}^{N} \sum_{k=1}^{L} (z_k^i)^2(1 - z_k^i)^2$ and $\mathcal{L}_{\text{mask\_cond}} = \sum_{i=0}^{N} \sum_{k=1}^{L} (m_k^i)^2(1 - m_k^i)^2$. The final loss will be:

$$\mathcal{L}_{\text{final}} = \mathcal{L}_{\text{adr}} + \delta\mathcal{L}_{\text{length}} + \gamma\mathcal{L}_{\text{Cat}} + \zeta\mathcal{L}_{\text{code\_cond}} + \mu\mathcal{L}_{\text{mask\_cond}}.$$ (5)

Note that for satisfying binary constraints, we can adopt other approaches. For instance, we can omit the requirement for this hyperparameter by using binary neural networks and an STE (straight-through estimator) [62]. Another approach would be to benefit from Boltzmann machines [63] to have a binary code. Having defined our theoretical objective, we are now ready to make it operational.

## 4 InfoSieve: Self-supervised Code Extraction

In this section, using the equations from Section 3, we devise a framework to extract category codes. Note that as Theorem 1 indicates, when we train the model to extract the category codes instead of the categories themselves, we make the model learn the underlying category tree. The model must ensure that in its implicit category tree, there is a node for each category whose descendants all share the same category while there are no non-descendants of this node with the same category.

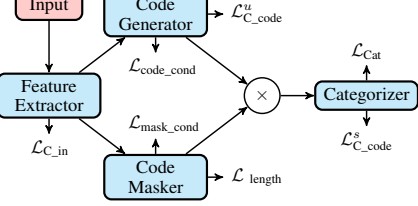

The overall framework of our model, named InfoSieve, is depicted in Figure 3. We first extract an embedding using the contrastive loss used by [27]. Then our *Code Generator* uses this embedding to generate binary codes, while our *Code Masker* learns a mask based on these embeddings to minimize the code length. Ultimately, the *Categorizer* uses this truncated code to discern ground-truth categories. In the next sections, we explain each component in more detail.

Figure 3: **InfoSieve framework.** *Feature Extractor* extracts an embedding by minimizing contrastive loss $\mathcal{L}_{\text{C\_in}}$. The *Code Generator* uses these input embeddings to find category codes. The *Code Masker* simultaneously learns masks that minimize the code length with $\mathcal{L}_{\text{length}}$. Finally, truncated category codes are used to minimize a contrastive loss for category codes while also predicting the seen categories by minimizing $\mathcal{L}_{\text{Cat}}$.

## 4.1 Contrastive Learning of Code and Input

One of the advantages of contrastive learning is to find a representation that maximizes the mutual information with the input [64]. For input $X^i$, let's show the hidden representation learning with $Z^i$, which is learned contrastively by minimizing the InfoNCE loss. Van den Oord et al. [64] showed that minimizing InfoNCE loss increases a lower bound for mutual information. Hence, contrastive learning with the InfoNCE loss can be a suitable choice for minimizing the $\mathcal{L}_{\text{adr}}$ in Equation 1. We will use this to our advantage on two different levels. Let's consider that $Z^i$ has dimension $d$, and each latent variable $z^i_k$ can take up $n$ different values. The complexity of the feature space for this latent variable would be $\mathcal{O}(n^d)$, then as we show in supplemental, the number of structurally different binary trees for this feature space will be $\mathcal{O}(4^{n^d})$. So minimizing $n$ and $d$ will be the most effective way to limit the number of possible binary trees. Since our model and the amount of training data is bounded, we must minimize the possible search space while providing reasonable performance. At the same time, the input feature space $X^i$ with $N$ possible values and dimension $D$ has $\mathcal{O}(N^D)$ possible states. To cover it completely, we can not arbitrarily decrease $d$ and $n$. Note that for a nearly continuous function $N \to \infty$, the probability of a random discrete tree fully covering this space would be near zero. To make the best of both worlds, we consider different levels of complexity of latent variables, each focusing on one of these goals.

**Minimizing Contrastive Loss on the Inputs.** Similar to [27], we use this unsupervised contrastive loss to maximize the mutual information between input $X^i$ and the extracted latent embedding $Z^i$. Akin to [27], we also benefit from the supervised contrastive learning signal for the members of a particular category. Let's assume that the number of categories in the entire dataset is $\mathcal{C}$. Different members of a category can be seen as different views of that category, analogous to unsupervised contrastive loss. Hence, they combine these unsupervised contrastive loss or $\mathcal{L}^u_{\text{C\_in}}$ and its supervised counterpart, $\mathcal{L}^s_{\text{C\_in}}$ with a coefficient $\lambda$, which we call $\lambda_{\text{in}}$ in a following manner:

$$\mathcal{L}_{C\_in} = (1 - \lambda_{\text{in}})\mathcal{L}^u_{\text{C\_in}} + \lambda_{\text{in}}\mathcal{L}^s_{\text{C\_in}} \tag{6}$$

For covering the input space, $X^i$, the loss function from Equation 6 is more suited if we consider both input and latent features as approximately continuous. We have shown this loss by $\mathcal{L}_{\text{C\_in}}$ in Figure 3.

**Minimizing Contrastive Loss on the Codes.** In order to also facilitate finding the binary tree, we map the latent feature extracted from the previous section to a binary code with minimum length. Hence we effectively decrease $n$ to 2 while actively minimizing $d$. Furthermore, we extract a code by making the value of each bit in this code correspond to its sequential position in a binary number, i.e., each digit has twice the value of the digit immediately to its right. This ensures the model treats this code as a binary tree coding with its root in the first digit. We consider unsupervised contrastive learning for raw binary digits $\mathcal{L}^u_{\text{C\_code}}$ and supervised variants of this loss for the extracted code, which we will show by and $\mathcal{L}^s_{\text{C\_code}}$, the total contrastive loss for the binary embedding is defined as:

$$\mathcal{L}_{\text{C\_code}} = (1 - \lambda_{\text{code}})\mathcal{L}^u_{\text{C\_code}} + \lambda_{\text{code}}\mathcal{L}^s_{\text{C\_code}} \tag{7}$$

In summary, the loss from Equation 6 finds a tree compatible with the input, while the loss from Equation 7 learns an implicit tree in compliance with categories. Then we consider $\mathcal{L}_{\text{adr}}$ as their combination:

$$\mathcal{L}_{\text{adr}} = \alpha\mathcal{L}_{\text{C\_in}} + \beta\mathcal{L}_{\text{C\_code}} \tag{8}$$

## 4.2 Minimizing Code Length

As discussed in Section 4.1, we need to decrease the feature space complexity by using minimum length codes to reduce the search space for finding the implicit category binary tree. In addition, using Shannon's mutual information as an approximate substitute for algorithmic mutual information necessitates minimizing the sequence length. We must simultaneously learn category codes and their optimal length. Since each of these optimizations depends on the optimal solution of the other one, we use two different blocks to solve them at the same time.

**Code Generator Block.** In Figure 3, the *Code Generator* block uses the extracted embeddings to generate binary category codes. At this stage, we consider a fixed length for these binary codes. The output of this stage is used for unsupervised contrastive learning on the codes in Equation 7. We also use $\mathcal{L}_{\text{code\_cond}}$ to enforce the digits of the codes to be a decent approximation of binary values.

**Code Masker Block.** For this block, we use the $\mathcal{L}_{\text{mask\_cond}}$ to ensure the binary constraint of the outputs. In addition, to control the length of the code or, in other words, the sequence of 1s at the beginning, we use $\mathcal{L}_{\text{length}}$ in Equation 2.

Table 1: **Ablation study on the effectiveness of each loss function.** Accuracy score on the CUB dataset is reported. This table indicates each component's preference for novel or known categories. In the first row of the table, differences in results compared with [27] can be attributed to specific implementation details, which are elaborated in section B.2 in the supplemental.

| $\mathcal{L}_{\text{C\_in}}$ | $\mathcal{L}_{\text{C\_code}}$ | $\mathcal{L}_{\text{code\_cond}}$ | $\mathcal{L}_{\text{length}}$ | $\mathcal{L}_{\text{mask\_cond}}$ | $\mathcal{L}_{\text{Cat}}$ | All | Known | Novel |
|:---:|:---:|:---:|:---:|:---:|:---:|:---:|:---:|:---:|
| ✓ | ✗ | ✗ | ✗ | ✗ | ✗ | 66.8 | 78.1 | 61.1 |
| ✓ | ✓ | ✗ | ✗ | ✗ | ✗ | 67.7 | 75.7 | 63.7 |
| ✓ | ✓ | ✓ | ✗ | ✗ | ✗ | 68.5 | 77.5 | 64.0 |
| ✓ | ✓ | ✓ | ✓ | ✗ | ✗ | 68.4 | 76.1 | 64.5 |
| ✓ | ✓ | ✓ | ✓ | ✓ | ✗ | 67.8 | 78.7 | 62.3 |
| ✓ | ✗ | ✗ | ✗ | ✗ | ✓ | 68.2 | 76.4 | 64.1 |
| ✓ | ✓ | ✓ | ✓ | ✓ | ✓ | 69.4 | 77.9 | 65.2 |

## 4.3 Aligning Codes using Supervision Signals

In the final block of the framework, we convert category codes from the *Code Generator* to a binary number based on the digit positions in the sequence. To truncate this code, we do a Hadamard multiplication for this number by the mask generated by the *Code Masker*. We use these truncated codes for supervised contrastive learning on the codes. Finally, we feed these codes to the *Catgorizer* block to predict the labels directly. We believe relying solely on contrastive supervision prevents the model from benefiting from learning discriminative features early on to speed up training.

# 5 Experiments

## 5.1 Experimental Setup

**Eight Datasets.** We evaluate our model on three coarse-grained datasets CIFAR10/100 [65] and ImageNet-100 [66] and four fine-grained datasets: CUB-200 [67], Aircraft [68], SCars [69] and Oxford-Pet [70]. Finally, we use the challenging Herbarium19 [71] dataset, which is fine-grained and long-tailed. To acquire the train and test splits, we follow [27]. We subsample the training dataset in a ratio of $50\%$ of known categories at the train and all samples of unknown categories. For all datasets except CIFAR100, we consider $50\%$ of the categories as known categories at training time. For CIFAR100, $80\%$ of the categories are known during training time, as in [27]. A summary of dataset statistics and their train test splits is shown in the supplemental.

**Implementation Details.** Following [27], we use ViT-B/16 as our backbone, which is pre-trained by DINO [72] on unlabelled ImageNet 1K [4]. Unless otherwise specified, we use 200 epochs and batch size of 128 for training. We present the complete implementation details in the supplemental. Our code is available at: `https://github.com/SarahRastegar/InfoSieve`.

**Evaluation Metrics.** We use semi-supervised $k$-means proposed by [27] to cluster the predicted embeddings. Then, the Hungarian algorithm [73] solves the optimal assignment of emerged clusters to their ground truth labels. We report the accuracy of the model's predictions on *All*, *Known*, and *Novel* categories. Accuracy on *All* is calculated using the whole unlabelled train set, consisting of known and unknown categories. For *Known*, we only consider the samples with labels known during training. Finally, for *Novel*, we consider samples from the unlabelled categories at train time.

## 5.2 Ablative studies

We investigate each model's component contribution to the overall performance of the model and the effect of each hyperparameter. Further ablations can be found in the supplemental.

**Effect of Each Component.** We first examine the effect of each component using the CUB dataset. A fine-grained dataset like CUB requires the model to distinguish between the semantic nuances of each category. Table 1 shows the effect of each loss component for the CUB dataset. As we can see from this table, $\mathcal{L}_{\text{C\_code}}$ and $\mathcal{L}_{\text{length}}$ have the most positive effect on novel categories while affecting known categories negatively. Utilizing $\mathcal{L}_{\text{code\_cond}}$ to enforce binary constraints on the embedding enhances performance for both novel and known categories. Conversely, applying $\mathcal{L}_{\text{mask\_cond}}$ boosts performance for known categories while detrimentally impacting novel categories. This is aligned with Theorem 1 and the definition of the category because these losses have opposite goals. As discussed, minimizing the search space helps the model find the implicit category tree faster. $\mathcal{L}_{\text{C\_code}}$ and $\mathcal{L}_{\text{length}}$ try to achieve this by mapping the information to a smaller feature space, while condition

Table 2: **Hyperparameter anlysis.** This table indicates the effect on the accuracy score of each hyperparameter on the CUB dataset for novel or known categories.

<table>
<tr><td colspan="4" align="center">**(a) Effect of Code Constraint Coefficient**</td></tr>
<tr><td>**Code Constraint Coef**</td><td>**All**</td><td>**Known**</td><td>**Novel**</td></tr>
<tr><td>$\zeta = 0.01$</td><td>69.4</td><td>**77.9**</td><td>65.2</td></tr>
<tr><td>$\zeta = 0.1$</td><td>69.1</td><td>76.1</td><td>65.6</td></tr>
<tr><td>$\zeta = 1$</td><td>**69.9**</td><td>76.1</td><td>**66.8**</td></tr>
<tr><td>$\zeta = 2$</td><td>69.5</td><td>75.5</td><td>66.5</td></tr>
</table>

<table>
<tr><td colspan="4" align="center">**(b) Effect of Mask Constraint Coefficient**</td></tr>
<tr><td>**Mask Constraint Coef**</td><td>**All**</td><td>**Known**</td><td>**Novel**</td></tr>
<tr><td>$\mu = 0.01$</td><td>69.4</td><td>77.9</td><td>65.2</td></tr>
<tr><td>$\mu = 0.1$</td><td>67.2</td><td>74.1</td><td>63.8</td></tr>
<tr><td>$\mu = 1$</td><td>69.3</td><td>76.0</td><td>65.9</td></tr>
<tr><td>$\mu = 2$</td><td>**70.6**</td><td>**79.3**</td><td>**66.3**</td></tr>
</table>

<table>
<tr><td colspan="4" align="center">**(c) Effect of Code Contrastive Coefficient**</td></tr>
<tr><td>**Code Contrastive Coef**</td><td>**All**</td><td>**Known**</td><td>**Novel**</td></tr>
<tr><td>$\beta = 0.01$</td><td>68.6</td><td>77.0</td><td>64.5</td></tr>
<tr><td>$\beta = 0.1$</td><td>**69.9**</td><td>76.3</td><td>**66.7**</td></tr>
<tr><td>$\beta = 1$</td><td>69.4</td><td>**77.9**</td><td>65.2</td></tr>
<tr><td>$\beta = 2$</td><td>68.6</td><td>**77.9**</td><td>64.0</td></tr>
</table>

<table>
<tr><td colspan="4" align="center">**(d) Effect of Code Length Coefficient**</td></tr>
<tr><td>**Code Cut Length Coef**</td><td>**All**</td><td>**Known**</td><td>**Novel**</td></tr>
<tr><td>$\delta = 0.01$</td><td>69.4</td><td>77.0</td><td>65.6</td></tr>
<tr><td>$\delta = 0.1$</td><td>69.4</td><td>**77.9**</td><td>65.2</td></tr>
<tr><td>$\delta = 1$</td><td>68.8</td><td>75.8</td><td>65.2</td></tr>
<tr><td>$\delta = 2$</td><td>**69.8**</td><td>76.9</td><td>**66.2**</td></tr>
</table>

losses solve this by pruning and discarding unnecessary information. Finally, $\mathcal{L}_{\text{Cat}}$ on its own has a destructive effect on known categories. One reason for this is the small size of the CUB dataset, which makes the model overfit on labeled data. However, when all losses are combined, their synergic effect helps perform well for both known and novel categories.

## 5.3 Hyperparameter Analysis

Our model has a few hyperparameters: code binary constraint ($\zeta$), mask binary constraint ($\mu$), code contrastive ($\beta$), code length ($\delta$), and code mapping ($\eta$). We examine the effect of each hyperparameter on the model's performance on the CUB dataset. Our default values for the hyperparameters are: code constraint coeff $\zeta$=0.01, mask constraint coeff $\mu$=0.01, code contrastive coeff $\beta$=1, code mapping coeff $\eta$=0.01 and code length coeff $\delta$=0.1.

**Code Binary Constraint.** This hyperparameter is introduced to satisfy the binary requirement of the code. Since we use tanh to create the binary vector, the coefficient only determines how fast the method satisfies the conditions. When codes $0$ and $1$ are stabilized, the hyperparameter effect will be diminished. However, we noticed that more significant coefficients somewhat affect the known accuracy. The effect of this hyperparameter for the CUB dataset is shown in Table 2 (a). We can see that the method is robust to the choice of this hyperparameter.

**Mask Binary Constraint.** For the mask constraint hyperparameter, we start from an all-one mask in our Lagrange multiplier approach. A more significant constraint translates to a longer category code. A higher coefficient is more useful since it better imposes the binary condition for the mask. The effect of different values of this hyperparameter for the CUB datasets is shown in Table 2 (b).

**Code Contrastive.** This loss maintains information about the input. From Table 2 (c), we observe that minimizing the information for a fine-grained dataset like CUB will lead to a better performance.

**Code Length.** Table 2 (d) reports our results for different values of code length hyperparameter. Since the code length is penalized exponentially, the code length hyperparameter's effect is not comparable to the exponential growth; hence, in the end, the model is not sensitive to this hyperparameter's value.

**Code Mapping.** Since current evaluation metrics rely on predefined categories, our category codes must be mapped to this scenario. This loss is not an essential part of the self-coding that our model learns, but it accelerates model training. The effect of this hyperparameter is shown in Table 3.

**Overall Parameter Analysis** One reason the model is not very sensitive to different hyperparameters is that our model consists of three separate parts: Code masker, Code Generator, and Categorizer. The only hyperparameters that affect all of these three parts directly are $\beta$, the code contrastive coefficient, and $\delta$, the code length coefficient. Hence, these hyperparameters affect our model's performance more.

Table 3: **Effect of Code Mapping.** Accuracy scores on the CUB dataset for novel and known categories.

| Code | CUB | | |
| --- | --- | --- | --- |
| | **All** | **Known** | **Novel** |
| $\eta = 0.01$ | **69.4** | **77.9** | 65.2 |
| $\eta = 0.1$ | 68.4 | 75.7 | 64.8 |
| $\eta = 1$ | 68.8 | 75.7 | **65.3** |
| $\eta = 2$ | 68.9 | 77.5 | 64.6 |

Table 4: **Comparison on fine-grained image recognition datasets.** Accuracy score for the first three methods is reported from [27] and for ORCA from [28]. Bold and underlined numbers, respectively, show the best and second-best accuracies. Our method has superior performance for the three experimental settings (*All*, *Known*, and *Novel*). This table shows that our method is especially well suited to fine-grained settings.

| Method | CUB-200 | | | FGVC-Aircraft | | | Stanford-Cars | | | Oxford-IIIT Pet | | | Herbarium-19 | | |
|---|---|---|---|---|---|---|---|---|---|---|---|---|---|---|---|
| | All | Known | Novel | All | Known | Novel | All | Known | Novel | All | Known | Novel | All | Known | Novel |
| k-means [74] | 34.3 | 38.9 | 32.1 | 12.9 | 12.9 | 12.8 | 12.8 | 10.6 | 13.8 | 77.1 | 70.1 | 80.7 | 13.0 | 12.2 | 13.4 |
| RankStats+ [75] | 33.3 | 51.6 | 24.2 | 26.9 | 36.4 | 22.2 | 28.3 | 61.8 | 12.1 | - | - | - | 27.9 | 55.8 | 12.8 |
| UNO+ [76] | 35.1 | 49.0 | 28.1 | 40.3 | 56.4 | 32.2 | 35.5 | 70.5 | 18.6 | - | - | - | 28.3 | 53.7 | 14.7 |
| ORCA [77] | 36.3 | 43.8 | 32.6 | 31.6 | 32.0 | 31.4 | 31.9 | 42.2 | 26.9 | - | - | - | 24.6 | 26.5 | 23.7 |
| GCD [27] | 51.3 | 56.6 | 48.7 | 45.0 | 41.1 | 46.9 | 39.0 | 57.6 | 29.9 | 80.2 | 85.1 | 77.6 | 35.4 | 51.0 | 27.0 |
| XCon [32] | 52.1 | 54.3 | 51.0 | 47.7 | 44.4 | 49.4 | 40.5 | 58.8 | 31.7 | 86.7 | _91.5_ | 84.1 | - | - | - |
| PromptCAL [28] | 62.9 | 64.4 | 62.1 | 52.2 | 52.2 | _52.3_ | 50.2 | 70.1 | 40.6 | - | - | - | - | - | - |
| DCCL [29] | _63.5_ | 60.8 | _64.9_ | - | - | - | 43.1 | 55.7 | 36.2 | _88.1_ | 88.2 | _88.0_ | - | - | - |
| SimGCD [31] | 60.3 | _65.6_ | 57.7 | _54.2_ | _59.1_ | 51.8 | _53.8_ | _71.9_ | _45.0_ | - | - | - | **44.0** | **58.0** | **36.4** |
| GPC [78] | 52.0 | 55.5 | 47.5 | 43.3 | 40.7 | 44.8 | 38.2 | 58.9 | 27.4 | - | - | - | - | - | - |
| InfoSieve | **69.4** | **77.9** | **65.2** | **56.3** | **63.7** | **52.5** | **55.7** | **74.8** | **46.4** | **91.8** | **92.6** | **91.3** | _41.0_ | _55.4_ | _33.2_ |

Table 5: **Comparison on coarse-grained image recognition datasets.** Accuracy for the first three methods from [27] and for ORCA from [28]. Bold and underlined numbers, respectively, show the best and second-best accuracies. While our method does not reach state-of-the-art for coarse-grained settings, it has a consistent performance for all three experimental settings (*All*, *Known*, *Novel*).

| Method | CIFAR-10 | | | CIFAR-100 | | | ImageNet-100 | | |
|---|---|---|---|---|---|---|---|---|---|
| | All | Known | Novel | All | Known | Novel | All | Known | Novel |
| k-means [74] | 83.6 | 85.7 | 82.5 | 52.0 | 52.2 | 50.8 | 72.7 | 75.5 | 71.3 |
| RankStats+ [75] | 46.8 | 19.2 | 60.5 | 58.2 | 77.6 | 19.3 | 37.1 | 61.6 | 24.8 |
| UNO+ [76] | 68.6 | **98.3** | 53.8 | 69.5 | 80.6 | 47.2 | 70.3 | **95.0** | 57.9 |
| ORCA [77] | 96.9 | 95.1 | 97.8 | 74.2 | 82.1 | 67.2 | 79.2 | 93.2 | 72.1 |
| GCD [27] | 91.5 | _97.9_ | 88.2 | 73.0 | 76.2 | 66.5 | 74.1 | 89.8 | 66.3 |
| XCon [32] | 96.0 | 97.3 | 95.4 | 74.2 | 81.2 | 60.3 | 77.6 | 93.5 | 69.7 |
| PromptCAL [28] | **97.9** | 96.6 | **98.5** | **81.2** | _84.2_ | _75.3_ | **83.1** | 92.7 | **78.3** |
| DCCL [29] | 96.3 | 96.5 | 96.9 | 75.3 | 76.8 | 70.2 | 80.5 | 90.5 | 76.2 |
| SimGCD [31] | _97.1_ | 95.1 | _98.1_ | _80.1_ | 81.2 | **77.8** | _83.0_ | 93.1 | _77.9_ |
| GPC [78] | 90.6 | 97.6 | 87.0 | 75.4 | **84.6** | 60.1 | 75.3 | 93.4 | 66.7 |
| InfoSieve | 94.8 | 97.7 | 93.4 | 78.3 | 82.2 | 70.5 | 80.5 | _93.8_ | 73.8 |

## 5.4 Comparison with State-of-the-Art

**Fine-grained Image Classification.** Fine-grained image datasets are a more realistic approach to the real world. In coarse-grained datasets, the model can use other visual cues to guess about the novelty of a category; fine-grained datasets require that the model distinguish subtle category-specific details. Table 4 summarizes our model's performance on the fine-grained datasets. As we can see from this table, our model has more robust and consistent results compared to other methods for fine-grained datasets. Herbarium 19, a long-tailed dataset, raises the stakes by having different frequencies for different categories, which is detrimental to most clustering approaches because of the extremely unbalanced cluster size. As Table 4 shows, our model can distinguish different categories even from a few examples and is robust to frequency imbalance.

**Coarse-grained Image Classification.** Our method is well suited for datasets with more categories and fine distinctions. Nevertheless, we also evaluate our model on three coarse-grained datasets, namely CIFAR10 and CIFAR100 [65] and ImageNet-100 [66]. Table 5 compares our results against state-of-the-art generalized category discovery methods. As we can see from this table, our method performs consistently well on both known and novel datasets. For instance, while UNO+ shows the highest accuracy on the *Known* categories of CIFAR-10, this is at the expense of performance degradation on the *Novel* categories. The same observations can be seen on ImageNet-100. Table 5 shows that our method consistently performs competitively for both novel and known categories. Based on our theory, the smaller CIFAR10/100 and ImageNet-100 improvement is predictable. For CIFAR 10, the depth of the implicit tree is 4; hence, the number of implicit possible binary trees with this limited depth is smaller, meaning finding a good approximation for the implicit category tree can be achieved by other models. However, as the depth of this tree increases, our model can still find the aforementioned tree; hence, we see more improvement for fine-grained data.

### 5.5 Qulitative results

In supplemental, we explain how to extract the implicit tree that is learned by our model. For instance, Figure 2 shows this extracted tree structure on the CUB dataset. We see that our method can extract some hierarchical structure in the data. In the leaves of this tree, we have "yellow sparrows," instances which are the descendants of a more abstract "sparrow" node; or in the path '00...1', we have a "blackbirds" node, which can encompass multiple blackbird species as its children nodes.

## 6 Related Works

**Novel Category Discovery** can be traced back to Han et al. [79], where knowledge from labeled data was used to infer the unknown categories. Following this work, Zhong et al. [80] solidified the novel class discovery as a new specific problem. The main goal of novel class discovery is to transfer the implicit category structure from the known categories to infer unknown categories [24, 26, 75, 76, 81, 82, 82–99]. Prior to the novel class discovery, the problem of encountering new classes at the test time was investigated by open-set recognition [16, 17, 20, 100]. However, the strategy of dealing with these new categories is different. In the open-set scenario, the model rejects the samples from novel categories, while novel class discovery aims to benefit from the vast knowledge of the unknown realm and infer the categories. However, the novel class discovery has a limiting assumption that test data only consists of novel categories. For a more realistic setting, *Generalized Category Discovery* considers both known and old categories at test time.

**Generalised Category Discovery** recently has been introduced by [27] and concurrently under the name *Open-world semi-supervised learning* by [77]. In this scenario, while the model should not lose its grasp on known categories, it must discover novel ones at test time. This adds an extra challenge because when we adapt the novel class discovery methods to this scenario, they are biased to either novel or known categories and miss the other group. There has been a recent surge of interest in generalized category discovery [28–31, 78, 101–110]. In this work, instead of viewing categories as an end, we investigated the fundamental question of how to conceptualize the *category* itself.

**Decision Tree Distillation.** The benefits of the hierarchical nature of categories have been investigated previously. Xiao [111] and Frosst and Hinton [112] used a decision tree in order to make the categorization interpretable. Adaptive neural trees proposed by [113] assimilate representation learning to its edges. Ji et al. [114] use attention binary neural tree to distinguish fine-grained categories by attending to the nuances of these categories. However, these methods need an explicit tree structure. In this work, we let the network extract this implicit tree on its own. This way, our model is also suitable when an explicit tree structure does not exist.

For a more comprehensive related work, see the supplemental.

## 7 Conclusion

This paper seeks to address the often neglected question of defining a *category*. While the concept of a *category* is readily accepted as a given in the exploration of novel categories and open-world settings, the inherent subjectivity of the term poses a challenge when optimizing deep networks, which are fundamentally mathematical frameworks. To circumvent this issue, we put forth a mathematical solution to extract category codes, replacing the simplistic one-hot categorical encoding. Further, we introduce a novel framework capable of uncovering unknown categories during inference and iteratively updating its environmental representation based on newly acquired knowledge. These category codes also prove beneficial in handling fine-grained categorization, where attention to nuanced differences between categories is paramount. By illuminating the limitations of the subjective notion of *category* and recasting them within a mathematical framework, we hope to catalyze the development of less human-dependent models.

## 8 Limitations

Despite the endeavors made in this work, there are several notable limitations. Foremost is our assumption of an implicit hierarchical tree underlying categorization. Furthermore, our optimization solution may not align perfectly with the actual implicit hierarchy, as it only satisfies the necessary conditions outlined in Theorem 1. Lastly, akin to other approaches in the generalized category discovery literature, we operate under the assumption that we have access to unlabeled data from unknown categories during training. This assumption, while convenient for model development, may not always hold true in real-world applications.

## Acknowledgments and Disclosure of Funding

This work is part of the project Real-Time Video Surveillance Search with project number 18038, which is (partly) financed by the Dutch Research Council (NWO) domain Applied and Engineering/Sciences (TTW). Special thanks to Dr. Dennis Koelma and Dr. Efstratios Gavves for their valuable insights about conceptualizing the *category* to alleviate generalized category discovery.

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

# A Theory

## A.1 Notation and Definitions

Let us first formalize our notation and definition for the rest of the section. Some definitions might overlap with the notations in the main paper. However, we repeat them here for ease of access.

**Probabilistic Notations.** We denote the input random variable with $X$ and the category random variable with $C$. The category code random variable, which we define as the embedding sequence of input $X^i$, is denoted by $z^i = z_1^i z_2^i \cdots z_L^i$, in which superscript $i$ shows the $i$th sample, while subscript $L$ shows the digit position in the code sequence.

**Coding Notations.** Let $\mathcal{C}$ be a countable set, we use $\mathcal{C}^*$ to show all possible finite sequences using the members of this set. For instance: $\{0,1\}^* = \{\epsilon, 0, 1, 00, 01, 10, 11, \cdots\}$ in which $\epsilon$ is empty word. The length of each sequence z, which we show with $l(\mathrm{z})$, equals the number of digits present in that sequence. For instance, for the sequence $l(01010) = 5$.

**Shannon Information Theory Notations.** We denote the *Shannon entropy* or *entropy* of the random variable $X$ with $H(X)$. It measures the randomness of values of $X$ when we only have knowledge about its distribution $P$. It also measures the minimum number of bits required on average to transmit or encode the values drawn from this probability distribution [50, 51]. The *conditional entropy* of a random variable $X$ given random variable $Z$ is shown by $H(X|Z)$, which states the amount of randomness we expect to see from $X$ after observing $Z$. In addition, $I(X; Z)$ indicates the *mutual information* between random variables $X$ and $Z$ [50, 51], which measures the amount of *information* we can obtain for one random variable by observing the other one. Note that contrary to $H(X|Z)$, mutual information is *symmetric*.

**Algorithmic Information Theory Notations.** Similar to Shannon's information theory, *Kolmogorov Complexity* or *Algorithmic Information Theory*[59–61] measures the shortest length to describe an object. Their difference is that Shannon's information considers that the objects can be described by the characteristic of the source that produces them, but *Kolmogorov Complexity* considers that the description of each object in isolation can be used to describe it with minimum length. For example, a binary string consisting of one thousand zeros might be assigned a code based on the underlying distribution it has been drawn from. However, *Kolmogorov Complexity* shows that we can encode this particular observation by transforming a description such as `"print 0 for 1000 times"`. The analogon to entropy is called *complexity* $K(\mathrm{x})$, which specifies the minimum length of a sequence that can *specify* output for a particular system. We denote the *algorithmic mutual information* for sequences x and z with $I_{alg}(\mathrm{x}:\mathrm{z})$, which specifies how much information about sequence x we can obtain by observing sequence z.

## A.2 Maximizing the Algorithmic Mutual Information

Let's consider data space $\mathcal{D}=\{X^i, C^i : i \in \{1, \cdots N\}\}$ where $X$s are inputs and $C$s are the corresponding category labels.

**Lemma 2** *For each category c and for $X^i$ with $C^i$=c, we can find a binary decision tree $\mathcal{T}_c$ that starting from its root, reaches each $X^i$ by following the decision tree path. Based on this path, we assign code $c(X^i)=c_1^i c_2^i \cdots c_M^i$ to each $X^i$ to uniquely define and retrieve it from the tree.*

**Proof of Lemma 2.** Since the number of examples in the dataset is finite, we can enumerate samples of category $c$ with any arbitrary coding. We then can replace these enumerations with their binary equivalent codes. We start from a root, and every time we encounter $1$ in digits of these codes, we add a right child node, and for $0$, we add a left child node. We then continue from the child node until we reach the code's end. Since the number of samples with category $c$ is limited, this process should terminate. On the other hand, since the binary codes for different samples are different, these paths are unique, and by the time we traverse a path from the root to a leaf node, we can identify the unique sample corresponding to that node. $\square$

As mentioned in the main paper, using this Lemma, we can find at least one supertree $\mathbf{T}$ for the whole data space that addresses all samples in which samples of the same category share a similar prefix. We can define a model that provides the binary code $z^i = z_1^i \cdots z_L^i$ for data input $X^i$ with category $c$ based on the path it takes in these eligible trees. We define these path encoding functions *valid encoding* as defined in Definition 2:

**Definition 2** *A valid encoding for input space $\mathcal{X}$ and category space $\mathcal{C}$ is defined as an encoding that uniquely identifies every $X^i \in \mathcal{X}$. At the same time, for each category $c \in \mathcal{C}$, it ensures that there is a sequence that is shared among all members of this category but no member out of the category.*

Since there is no condition on how to create these trees and their subtrees, many candidate trees can address the whole data space while preserving a similar prefix for the members of each category.

However, based on our inspirations for how the brain does categorization, we assume the ground truth underlying tree $\mathbf{T}$ has a minimum average length path from the root to each node. In other words, each sample $x$ has the shortest description code $z$ to describe that data point while maintaining its validity. If we use a model to learn this encoding, the optimal model tree should be isomorph to the underlying tree $\mathbf{T}$,

**Lemma 3** *For a learned binary code $\mathrm{z}^i$ to address input $X^i$, uniquely, if the decision tree of this encoding is optimal, it is isomorph to the underlying tree $T$.*

**Proof of Lemma 3.** Since the underlying tree has the minimum Kolmogorov complexity for each sample, we can extract the optimal lengths of each sample by traversing the tree. Evans and Lanoue [115] showed that a tree can be recovered from the sequence of lengths of the paths from the root to leaves to the level of isomorphism. Based on our assumption about the underlying tree $\mathbf{T}$, the optimal tree can not have a shorter length for any sample codes than the underlying tree. On the other hand, having longer codes contradicts its optimality. Hence the optimal tree should have similar path lengths to the underlying ground truth tree. Therefore, it is isomorphic to the underlying tree. $\square$

Since the optimal tree with the valid encoding $\tilde{z}$ is isomorph to the underlying tree, we will have the necessary conditions that Theorem 1 provides.

**Theorem 1** *For a learned binary code $\mathrm{z}^i$ to address input $x^i$, uniquely, if the decision tree of this encoding is isomorph to underlying tree $\mathbf{T}$, we will have the following necessary conditions:*

1. *$I_{alg}(\mathrm{z} : x) \geq I_{alg}(\tilde{z} : x) \quad \forall \tilde{z}, \tilde{z}$ is a valid encoding for $x$*

2. *$I_{alg}(\mathrm{z} : c) \geq I_{alg}(\tilde{z} : c) \quad \forall \tilde{z}, \tilde{z}$ is a valid encoding for $x$*

**Proof of Theorem 1.**

*Part one*: From the way $\mathbf{T}$ has been constructed, we know that $K(x|\mathbf{T}) \leq K(x|\mathcal{T})$ in which $\mathcal{T}$ is an arbitrary tree. From the complexity and mutual information properties, we also have $I_{\mathrm{alg}}(\mathrm{z} : x) = K(\mathrm{z}) - K(x|\mathrm{z})$ [116]. Since $\tilde{z}$ and z have isomorph tree structures, then $K(\tilde{z}) = K(\mathrm{z})$, hence: $I_{\mathrm{alg}}(\mathrm{z} : x) \geq I_{\mathrm{alg}}(\tilde{z} : x)$. $\square$

*Part two*: In any tree that is a valid encoding, all samples of a category should be the descendants of that node. Thus, the path length to corresponding nodes should be similar in both trees. Otherwise, the length of the path to all samples of this category will not be optimal. We can use the same logic and deduce that the subtree with the category nodes as its leaves would be isomorph for both embeddings. Let's denote the path from the root to category nodes with $\mathrm{z}_c$ and from the category node to its corresponding samples with $\mathrm{z}_x$. If we assume these two paths can be considered independent, we will have $K(x) = K(\mathrm{z}_c\mathrm{z}_x) = K(\mathrm{z}_c) + K(\mathrm{z}_x)$, which indicates that minimizing $K(x)$ in the tree implies that $K(c)$ also should be minimized. By applying the same logic as part one, we can deduce that $I_{\mathrm{alg}}(\mathrm{z} : c) \geq I_{\mathrm{alg}}(\tilde{z} : c)$. $\square$

### A.2.1 Shannon Mutual Information Approximation

Optimization in Theorem 1 is generally not computable [58–61]. However, We can approximate these requirements using Shannon mutual information instead. Let's consider two functions $f$ and $g$, such that both are $\{0,1\}^* \to \mathbb{R}$. For these functions, $f \stackrel{+}{<} g$ means that there exists a constant $\kappa$, such that $f \leq g + c$, when both $f \stackrel{+}{<} g$ and $g \stackrel{+}{<} f$ hold, then $f \stackrel{+}{=} g$ [116].

**Theorem 2** *[116] Let $P$ be a computable probability distribution on $\{0,1\}^* \times \{0,1\}^*$. Then:*

$$I(X;Z) - K(P) \stackrel{+}{<} \sum_x \sum_z p(x,z) I_{alg}(\mathrm{x} : \mathrm{z}) \stackrel{+}{<} I(X;Z) + 2K(P) \qquad (9)$$

This theorem states that the expected value of algorithmic mutual information is close to its probabilistic counterpart. This means that if we maximize the Shannon information, we also approximately maximize the algorithmic information and vice versa.

Since Shannon entropy does not consider the inner regularity of the symbols it codes, to make each sequence meaningful from a probabilistic perspective, we convert each sequence to an equivalent random variable number by considering its binary digit representation. To this end, we consider $Z^i = \sum_{k=1}^{m} \frac{z_k^i}{2^k}$, which is a number between 0 and 1. Note that we can recover the sequence from the value of this random variable. Since the differences in the first bits affect the number more, for different error thresholds, Shannon's information will focus on the initial bits more. In dealing with real-world data, the first bits of encoding of a category sequence are more valuable than later ones due to the hierarchical nature of categories. Furthermore, with this tweak, we equip Shannon's model with a knowledge of different positions of digits in a sequence. To replace the first item of Theorem 1 by its equivalent Shannon mutual information, we must also ensure that z has the minimum length. For the moment, let's assume we know this length by the function $l(X^i) = l_i$. Instead of $Z^i$, we can consider its truncated form $Z_{l_i}^i = \sum_{k=1}^{l_i} \frac{z_k^i}{2^k}$. This term, which we call the address loss function, is defined as follows:

$$\mathcal{L}_{\text{adr}} = -\frac{1}{N} \sum_{i=0}^{N} I(X^i; Z_{l_i}^i) \quad s.t. \quad Z_{l_i}^i = \sum_{k=1}^{l_i} \frac{z_k^i}{2^k} \text{ and } \forall k, z_k^i \in \{0, 1\}. \tag{10}$$

We can approximate this optimization with a reconstruction or contrastive loss.

### A.2.2 Approximation with Reconstruction Loss

Let's approximate the maximization of the mutual information by minimizing the $\mathcal{L}_{MSE}$ of the reconstruction from the code z. Suppose that $D(X)$ is the decoder function, and it is a Lipschitz continuous function, which is a valid assumption for most deep networks with conventional activation functions [117]. We can find an upper bound for $\mathcal{L}_{MSE}$ using Lemma 3.

**Lemma 3** *Suppose that $D(X)$ is a Lipschitz continuous function with Lipschitz constant $\kappa$, then we will have the following upper bound for $\mathcal{L}_{MSE}$:*

$$\mathcal{L}_{MSE}(X) \leq \kappa \frac{1}{N} \sum_{i=0}^{N} 2^{-2l_i}$$

**Proof of Lemma 3**. Let's consider the $\mathcal{L}_{MSE}$ loss for the reconstruction $\hat{X}^i$ from the code $Z^i$. We denote reconstruction from the truncated category code $Z_{l_i}^i$ with $\hat{X}_{l_i}^i$.

$$\mathcal{L}_{MSE}(X) = \frac{1}{N} \sum_{i=0}^{N} \| \hat{X}_{l_i}^i - X^i \|^2$$

If we expand this loss, we will have the following:

$$\mathcal{L}_{MSE}(X) = \frac{1}{N} \sum_{i=0}^{N} \| D(Z_{L(X^i)}^i) - X^i \|^2$$

$$= \frac{1}{N} \sum_{i=0}^{N} \| D(\sum_{k=0}^{l_i} \frac{z_k^i}{2^k}) - X^i \|^2$$

Let's assume the optimal model can reconstruct $X^i$ using the entire code length $Z^i$, i.e. $X^i = D(\sum_{k=0}^{m} \frac{z_k^i}{2^k})$. Now let's replace this in the equation:

$$\mathcal{L}_{MSE}(X) = \frac{1}{N} \sum_{i=0}^{N} \| D(\sum_{k=0}^{l_i} \frac{z_k^i}{2^k}) - D(\sum_{k=0}^{m} \frac{z_k^i}{2^k}) \|^2$$

Given that $D(X)$ is a Lipschitz continuous function with the Lipschitz constant $\kappa$, then we will have the following:

$$
\begin{aligned}
\mathcal{L}_{MSE}(X) \leq & \kappa \frac{1}{N} \sum_{i=0}^{N} \| \sum_{k=0}^{l_i} \frac{z_k^i}{2^k} - \sum_{k=0}^{m} \frac{z_k^i}{2^k} \|^2 \\
\leq & \kappa \frac{1}{N} \sum_{i=0}^{N} \| 2^{-l_i} \|^2 \\
= & \kappa \frac{1}{N} \sum_{i=0}^{N} 2^{-2l_i} \qquad \square
\end{aligned}
$$

Lemma 3 indicates that to minimize the upper bound on $\mathcal{L}_{MSE}$, we should aim for codes with maximum length, which can also be seen intuitively. The more length of latent code we preserve, the more accurate the reconstruction would be. This is in direct contrast with the length minimization of the algorithmic mutual information. So, the tradeoff between these two objectives defines the optimal final length of the category codes.

### A.2.3 Approximation with Contrastive Loss

One of the advantages of contrastive learning is to find a representation that maximizes the mutual information with the input [64]. More precisely, if for input $X^i$, we show the hidden representation learning $Z^i$, that is learned contrastively by minimizing the InfoNCE loss, [64] showed that the following lower bound on mutual information exists:

$$
I(X^i; Z^i) \geq \log(N) - \mathcal{L}_N. \tag{11}
$$

Here, $\mathcal{L}_N$ is the InfoNCE loss, and $N$ indicates the sample size consisting of one positive and $N-1$ negative samples. Equation 11 shows that contrastive learning with the InfoNCE loss can be a suitable choice for minimizing the $\mathcal{L}_{adr}$ in Equation 10. We will use this to our advantage on two different levels. Let's consider that $Z^i$ has dimension $d$, and each latent variable $z_k^i$ can take up $n$ different values. The complexity of the feature space for this latent variable would be $\mathcal{O}(n^d)$, then the number of structurally different binary trees for this feature space would be $\mathcal{O}(C_{n^d})$, in which $C_i$ is the $i$th Catalan number, which asymptotically grows as $\mathcal{O}(4^i)$. Hence the number of possible binary taxonomies for the categories will be $\mathcal{O}(4^{n^d})$. So minimizing $n$ and, to a lesser degree, $d$, will be the most effective way to limit the number of possible binary trees. Since our model and the amount of training data is bounded, we must minimize the possible search space while still providing reasonable performance. On the other hand, the input feature space $X^i$ with $N$ possible values and dimension $D$ has $\mathcal{O}(N^D)$ possible states, and to cover it completely, we can not arbitrarily decrease $d$ and $n$. Note that for a nearly continuous function $N \to \infty$, the probability of a random discrete tree to fully covering this space would be near zero.

### A.3 Category Code Length Minimization

In the main paper, we indicate the code length loss $\mathcal{L}_{length}$, which we define as $\mathcal{L}_{\text{length}} = \frac{1}{N} \sum_{i=0}^{N} l_i$. To minimize this loss, we define a binary mask sequence $m^i = m_1^i m_2^i \cdots m_L^i$ to simulate the subscript property of $l_i$. We discussed minimizing the $L_p$ Norm for the weighted version of the mask, which we denote with $\bar{m}^i = (m_1^i 2^1)(m_2^i 2^2) \cdots (m_L^i 2^L)$. This will ensure the requirements because adding one extra bit has an equivalent loss of all previous bits.

$$
\mathcal{L}_{\text{length}} \approx \frac{1}{N} \sum_{i=0}^{N} \| \bar{m}^i \|_p . \tag{12}
$$

**Lemma 4** *Consider the weighted mask $\bar{m} = (m_1 2^1)(m_2 2^2) \cdots (m_L 2^L)$ where $m_j s$ are $0$ or $1$. Consider the norm $\| \bar{m} \|_p$ where $p \geq 1$, the rightmost $1$ digit contributes more to the norm than the entire left sequence.*

**Proof of Lemma 4.** Let's consider the loss function for mask $\bar{\mathrm{m}}=(m_1 2^1)(m_2 2^2)\cdots(m_L 2^L)$ and let's denote the rightmost 1 index, with $k$, for simplicity we consider the $\| \bar{\mathrm{m}} \|_p^p$:

$$\| \bar{\mathrm{m}} \|_p^p = \sum_{j=0}^{L}(m_j 2^j)^p = \sum_{j=0}^{k-1}(m_j 2^j)^p + (m_k 2^k)^p + \sum_{j=k+1}^{L}(m_j 2^j)^p$$

given that $m_j = 0, \forall j > k$ and $m_k = 1$, we will have:

$$\| \bar{\mathrm{m}} \|_p^p == \sum_{j=0}^{k-1}(m_j 2^j)^p + 2^{kp} + 0$$

now let's compare the two subparts of the right-hand side with each other:

$$\sum_{j=0}^{k-1}(m_j 2^j)^p \leq \sum_{j=0}^{k-1}(2^j)^p = \frac{2^{kp}-1}{2^p - 1} < 2^{kp} \qquad \square$$

Hence $\mathcal{L}_{Length}$ tries to minimize the position of the rightmost 1, simulating the cutting length subscript.

### A.3.1 Satisfying Binary Constraints.

In the main paper, we stated that we have two conditions, *Code Constraint:*$\forall z_k^i$, $z_k^i = 0$ *or* $z_k^i = 1$ and *Mask Constraint* $\forall m_k^i$, $m_k^i = 0$ *or* $m_k^i = 1$. We formulate each constraint in an equivalent Lagrangian function to make sure they are satisfied. For the binary code constraint we consider $f_{\text{code}}(z_k^i)=(z_k^i)(1 - z_k^i)=0$, which is only zero if $z_k^i=0$ or $z_k^i=1$. Similarly, for the binary mask constraint, we have $f_{\text{mask}}(m_k^i)=(m_k^i)(1-m_k^i)=0$. To ensure these constraints are satisfied, we optimize them with the Lagrangian function of the overall loss. Consider the Lagrangian function for $\mathcal{L}_{\text{total}}$,

$$\mathbf{L}(\mathcal{L}_{\text{total}}, \eta, \mu) = \mathcal{L}_{\text{total}} + \eta\mathcal{L}_{\text{code\_cond}} + \mu\mathcal{L}_{\text{mask\_cond}}$$

This lagrangian function ensures that constraints are satisfied for $\eta \to +\infty$ and $\mu \to +\infty$. Note that our method uses a tanh activation function which has been mapped between 0 and 1, to produce $m_k$ and $z_k$, so the conditions are always greater or equal to zero. For an unbounded output, we can consider the squared version of constraint functions to ensure that constraints will be satisfied. This shows how we reach the final unconstrained loss function in the paper.

## B Experiments

### B.1 Dataset Details

To acquire the train and test splits, we follow [27]. We subsample the training dataset in a ratio of 50% of known categories at the train and all samples of unknown categories. For all datasets except CIFAR100, we consider 50% of categories as known categories at training time. For CIFAR100 as in [27] 80% of the categories are known during training time. A summary of dataset statistics and their train test splits is shown in Table 6.

**CIFAR10/100**[65] are coarse-grained datasets consisting of general categories such as *car, ship, airplane, truck, horse, deer, cat, dog, frog* and *bird*.

**ImageNet-100** is a subset of 100 categories from the coarse-grained ImageNet [66] dataset.

**CUB** or the Caltech-UCSD Birds-200-2011 (CUB-200-2011) [67] is one of the most used datasets for fine-grained image recognition. It contains different bird species, which should be distinguished by relying on subtle details.

**FGVC-Aircraft** or Fine-Grained Visual Classification of Aircraft [68] dataset is another fine-grained dataset, which, instead of animals, relies on airplanes. This might be challenging for image recognition models since, in this dataset, structure changes with design.

**SCars** or Stanford Cars [69] is a fine-grained dataset of different brands of cars. This is challenging since the same brand of cars can look different from different angles or with different colors.

Table 6: **Statistics of datasets and their data splits for the generalized category discovery task.** The first three datasets are coarse-grained image classification datasets, while the next four are fine-grained datasets. The Herbarium19 dataset is both fine-grained and long-tailed.

| Dataset | Labelled | | Unlabelled | |
|---|---|---|---|---|
| | #Images | #Categories | #Images | #Categories |
| CIFAR-10 [65] | 12.5K | 5 | 37.5K | 10 |
| CIFAR-100 [65] | 20.0K | 80 | 30.0K | 100 |
| ImageNet-100 [66] | 31.9K | 50 | 95.3K | 100 |
| CUB-200 [67] | 1.5K | 100 | 4.5K | 200 |
| SCars [69] | 2.0K | 98 | 6.1K | 196 |
| Aircraft [68] | 3.4K | 50 | 6.6K | 100 |
| Oxford-Pet [70] | 0.9K | 19 | 2.7K | 37 |
| Herbarium19 [71] | 8.9K | 341 | 25.4K | 683 |

**Oxford-Pet** [70] is a fine-grained dataset of different species of cats and dogs. This is challenging since the amount of data is very limited in this dataset, which makes it prone to overfitting.

**Herbarium_19** [71] is a botanical research dataset about different types of plants. Due to its long-tailed alongside fine-grained nature, it is a challenging dataset for discovering novel categories.

## B.2 Implementation details

In this section, we provide our implementation details for each block separately. As mentioned in the main paper, the final loss function that we use to train the model is:

$$\mathcal{L}_{\text{final}} = \mathcal{L}_{\text{adr}} + \delta\mathcal{L}_{\text{length}} + \eta\mathcal{L}_{\text{Cat}} + \zeta\mathcal{L}_{\text{code\_cond}} + \mu\mathcal{L}_{\text{mask\_cond}}. \tag{13}$$

In which the loss $\mathcal{L}_{\text{adr}}$ is:

$$\mathcal{L}_{\text{adr}} = \alpha\mathcal{L}_{\text{C\_in}} + \beta\mathcal{L}_{\text{C\_code}}. \tag{14}$$

In this formula, $\mathcal{L}_{\text{C\_in}}$ is the loss function that [27] suggested, so we use the same hyperparameters as their defaults for this loss. Hence, we only expand on $\mathcal{L}_{\text{C\_code}}$:

$$\mathcal{L}_{\text{adr}} = \alpha\mathcal{L}_{\text{C\_in}} + \beta((1 - \lambda_{\text{code}})\mathcal{L}_{\text{C\_code}}^u + \lambda_{\text{code}}\mathcal{L}_{\text{C\_code}}^s). \tag{15}$$

In the scope of our experimentation, it was assumed by default that $\alpha=1$ and $\lambda_{\text{code}}=0.35$. The code generation process introduces a certain noise level, potentially leading to confusion in the model, particularly in fine-grained data. To mitigate this, we integrated a smoothing hyperparameter within our contrastive learning framework, aiming to balance the noise impact and avert excessive confidence in the generated code, for datasets such as CUB and Pet, the smoothing factor was set at 1, whereas for SCars, Aircraft, and Herb datasets, it was adjusted to 0.1. In contrast, we did not apply smoothing for generic datasets like CIFAR 10/100 and ImageNet, where label noise is less significant.

Furthermore, in dealing with fine-grained data, we opted to fine-tune the final two blocks of the DINO model. This approach differs from our strategy for generic datasets, where only the last block underwent fine-tuning. Additionally, we employed semi-supervised $k$-means at every epoch to derive pseudo-labels from unlabeled data. These pseudo-labels were then used in our supervised contrastive learning process as a supervisory signal. It is important to note that in supervised contrastive learning, the primary requirement is that paired samples belong to the same class, allowing us to disregard discrepancies between novel class pseudo-labels and their actual ground truth values. Furthermore, instead of cosine similarity for contrastive learning, we adopt Euclidean distance, a better approximation for the category problem. Finally, for balanced datasets, we use the balanced version of $k$-means for semi-supervised $k$-means.

**Code Generator.** To create this block, we use a fully connected network with GeLU activation functions [118]. Then, we apply a tanh activation function $\tanh(ax)$ in which $a$ is a hyperparameter showing the model's age. We expect that as the model's age increases or, in other words, in later epochs, the model will be more decisive because of sharper transitions from $0$ to $1$. Hence, we will have a stronger binary dichotomy for code values. Also, since contrastive learning makes the different samples as far as possible, this causes a problem for the Code Generator because the feature space will not smoothly transition from different samples of the same category, especially for fine-grained datasets. To alleviate this problem, we use a label smoothing hyperparameter in the contrastive objective to help make feature space smoother, which will require a smaller tree for encoding. Since the model should distinguish $0$s for the mask from $0$s of the code, we do not adjust the code generator to $0$ and $1$s and consider the $-1$ and $1$ values in practice.

**Code Masker.** The *Code Masker* block is a fully connected network with tanh activation functions at the end, which are adjusted to be $0$ and $1$s. We also consider the aging hyperparameter for the tanh activation function in the masking block. In the beginning, since codes are not learned, masking the embedding space might hamper its learning ability. To solve this, we start masker with all one's entries and gradually decrease it with epochs. Hence, the activation function that is applied to the masker would be $\tanh(x + \frac{1}{a+1})$, in which $a$ is the aging parameter. In practice, we observed that norm one is stable enough in this loss function while also truncating codes at a reasonable length. Since $\mathcal{L}_{\text{length}}$ grows exponentially with code length, it will mask most of the code. For fine-grained datasets, this could be detrimental for very similar categories. To alleviate this problem, instead of using $2$ as a positional base, we decrease it with each epoch to $2 - \frac{\text{epoch}}{N_{\text{epochs}}}$. So, at the end of training, the values of all positions are the same. This allows the model to encode more levels to the tree. Since we start with the base $2$, we are constructing the tree with a focus on nodes near the root at the start and to the leaves at the end of training.

**Categorizer.** We use a fully connected network for this block and train it with the one-hot encoding of the labeled samples. This module receives the truncated codes to predict the labeled data. This module cannot categorize labeled data if the masker truncates too much information. Hence, it creates error signals that prevent the masker from truncating too much. This part of the network is arbitrary, and we showed in ablations that we can ignore this module without supervision signals.

### B.3  Further Ablations

**Feature Space Visualization.** Figure 4 illustrates the tSNE visualizations for different embedding extracted from our model. While our model's features form separate clusters, our label embedding, which is the raw code feature before binarization, makes these clusters distinctive. After that, binary embedding enhances this separation while condensing the cluster by making samples of clusters closer to each other, which is evident for the bird cluster shown in yellow. Because of its $0$ or $1$ nature, semantic similarity will affect the binary embedding more than visual similarity. Finally, our code embedding, which assigns positional values to the extracted binary embedding, shows indirectly that to have the most efficient code, our model should span the code space as much as possible, which explains the porous nature of these clusters.

### B.4  Extracting the Implicit Tree from the Model

Suppose that the generated feature vector by the network for sample $X$ is $x_0 x_1 \cdots x_k$, where $k$ is the dimension of the code embedding or, equivalently, the depth of our implicit hierarchy tree. Using appropriate activation functions, we can assume that $x_i$ is binary. The unsupervised contrastive loss forces the model to make the associated code to each sample unique. So if $X'$ is not equivalent to $X$ or one of its augmentations, its code $x'_0 x'_1 \cdots x'_k$ will differ from the code assigned to $X$. For the supervised contrastive loss, instead of considering the code, we consider a sequence by assigning different positional values to each bit so the code $x_0 x_1 \cdots x_k$ can be considered as the binary number $0.x_0 x_1 \cdots x_k$. Then, the supervised contrastive loss aims to minimize the difference between these assigned binary numbers. This means our model learns to use the first digits for discriminative information while pushing the specific information about each sample to the last digits. Then, our masker learns to minimize the number of discriminative digits. Our Theorem states that, finally, the embedded tree that the model learns this way is a good approximation of the optimal tree. Ultimately, our model generates a code for each sample, and we consider each code as a binary tree traverse from the root to the leaf. Hence, the codes delineate our tree's structure and binary classification that

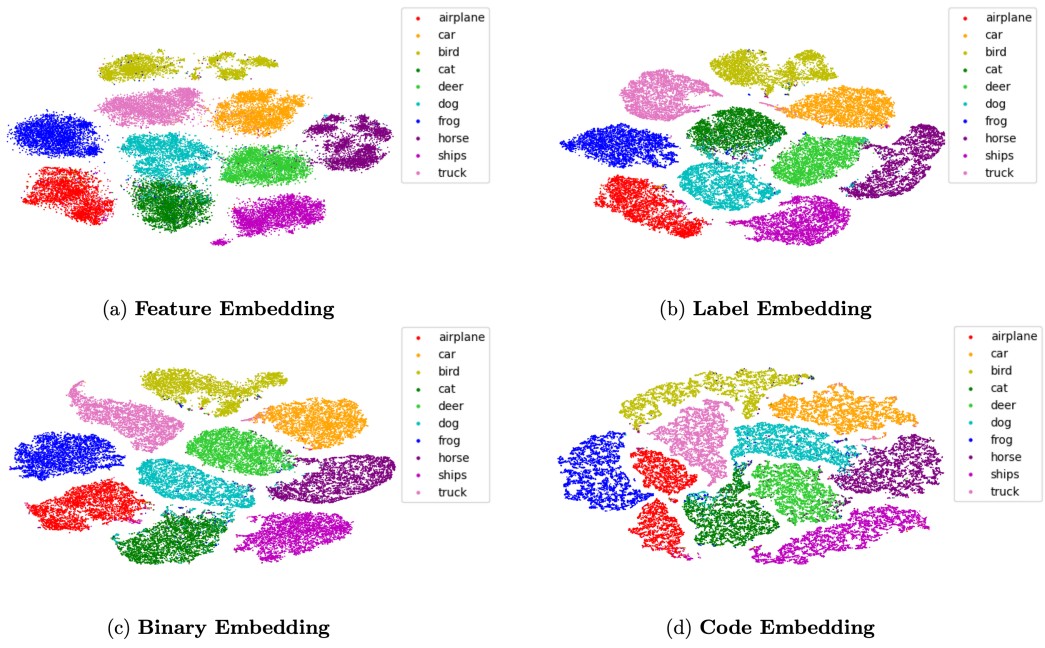

(a) **Feature Embedding**   (b) **Label Embedding**

(c) **Binary Embedding**   (d) **Code Embedding**

Figure 4: t-SNE plot for different embeddings in our model. **(a) Feature embedding.** The embedding after the projection head which is used by contrastive loss to maximize the representation information. **(b) Label embedding.** The embedding after generating code features is used by unsupervised contrastive loss for codes. **(c) Binary embedding.** The embedding by converting code features to a binary sequence using tanh activation functions and binary conditions. **(d) Code embedding.** The final truncated code which is generated by assigning positional values to the binary sequence and truncating the produced code using the masker network.

happens at each node. Since our approach enables the model to use the initial bits for supervised contrastive learning and the last bits for unsupervised contrastive learning, we can benefit from their synergic advantages while preventing them from interfering with each other.

## C   Related Works

### C.1   Open Set Recognition

The first sparks of the requirement for models that can handle real-world data were introduced by Scheirer et al. [20] and following works of [16, 18]. The first application of deep networks to address this problem was presented by OpenMax [17]. The main goal for open-set recognition is to distinguish *known* categories from each other while rejecting samples from *novel* categories. Hence many open-set methods rely on simulating this notion of *otherness*, either through large reconstruction errors [119, 120] distance from a set of prototypes[121–123] or by distinguishing the adversarially generated samples [124–127]. One of the shortcomings of open set recognition is that all new classes will be discarded.

### C.2   Novel Class Discovery

To overcome open set recognition shortcomings, *novel class discovery* aims to benefit from the vast knowledge of the unknown realm and infer the categories. It can be traced back to [79], where they used the knowledge from labeled data to infer the unknown categories. Following this work, [80] solidified the novel class discovery as a new specific problem. The main goal of novel class discovery is to transfer the implicit category structure from the known categories to infer unknown categories [24, 26, 75, 76, 81, 82, 82–99]. Despite this, the novel class discovery has a limiting assumption that test data only consists of novel categories.

## C.3 Generalized Category Discovery

For a more realistic setting, *Generalized Category Discovery* considers both known and old categories at the test time. This nascent problem was introduced by [27] and concurrently under the name *open-world semi-supervised learning* by [77]. In this scenario, while the model should not lose its grasp on old categories, it must discover novel categories in test time. This adds an extra challenge because when we adapt the novel class discovery methods to this scenario, they try to be biased to either novel or old categories and miss the other group. There has been a recent surge of interest in generalized category discovery [28–31, 78, 101–110]. In this work, instead of viewing categories as an end, we investigated the fundamental question of how to conceptualize *category* itself.

## C.4 Binary Tree Distillation

Benefiting from the hierarchical nature of categories has been investigated previously. Xiao [111] and Frosst and Hinton [112] used a decision tree in order to make the categorization interpretable and as a series of decisions. Adaptive neural trees proposed by [113] assimilate representation learning to its edges. Ji et al. [114] use attention binary neural tree to distinguish fine-grained categories by attending to the nuances of these categories. However, these methods need an explicit tree structure. In this work, we let the network extract this implicit tree on its own. This way, our model is also suitable when an explicit tree structure does not exist.

