# OpenReview forum: "Learn to Categorize or Categorize to Learn? Self-Coding for Generalized Category Discovery"
_NeurIPS.cc/2023/Conference — NeurIPS 2023 poster_

### Official Review · Reviewer_GHXK · 2023-07-03

**Soundness:** 3 good
**Presentation:** 3 good
**Contribution:** 2 fair
**Rating:** 5
**Confidence:** 3

**Summary:**

This paper introduces a novel perspective on categorization, framing it as an optimization problem with the goal of finding optimal solutions. The authors propose a self-supervised approach that can identify new categories during testing. They achieve this by assigning concise category codes to individual data instances, which effectively captures the inherent hierarchical structure found in real-world datasets.

**Strengths:**

1. The paper conceptualizes a category as a solution to an optimization problem, which is novel.
2. The paper is well-written, easy to follow
3. The proposed method is novel



**Weaknesses:**

1. The method is too complicated, with 5 hyperparameters in the overall loss functions, which is hard to tune, is it possible to make it automatically?
2. Some ablation studies are missing, I am curious about the effect on every hyperparameter in the loss function, how sensitive are they to the performance?
3. Some recent methods of comparison are missing.
* [1] PromptCAL: Contrastive Affinity Learning via Auxiliary Prompts for Generalized Novel Category Discovery (CVPR 2023)
* [2] Modeling Inter-Class and Intra-Class Constraints in Novel Class Discovery (CVPR 2023)
* [3] Generalized Category Discovery with Decoupled Prototypical Network (AAAI 2023)
* [4] Supervised Knowledge May Hurt Novel Class Discovery Performance (TMLR 2023)


**Questions:**

1. About the experiment setting, transductive or instructive?  the test set is included or excluded in the training process?
2. How to compute mutual information, MINE[1]?


[1]MINE: Mutual Information Neural Estimation


**Limitations:**

1. Complex Hyperparameter Tuning: The model's complexity with five hyperparameters in the loss functions makes tuning challenging. An automated tuning method might streamline this process.
2. Missing Ablation Studies: The absence of ablation studies limits understanding of individual hyperparameters' impact on performance. This analysis would contribute to model optimization.
3. Incomplete Comparison: The paper lacks comparison with some recent methods. Incorporating them can provide a more comprehensive evaluation.

---

> ### Author Rebuttal · Authors · 2023-08-09
>
> ***We thank the reviewer for finding our method and theory novel. We will use the provided references to enrich our paper both in related works and experimental analysis. Automatic hyperparameter tuning is an intriguing future work, we also provided the updated results vs. SoTA methods for stronger empirical proof.***
>
> **Complex Hyperparameter Tuning.** Since this paper tries to show that concept of learning categories as an arbitrary number can be detrimental to supervision, we suggested defining an optimization for extracting this inherent information from the data itself. The method proposed in the paper is one approach to this optimization. For instance, two condition losses can be replaced by using binary vectors from scratch. The main losses for the paper are length minimization and information maximization losses. Thus, only these hyperparameters are specific to our theory; other hyperparameters can differ based on how this optimization problem is approximated. Adopting an automatic hyperparameter tuning method can help make models more autonomous and interesting in future work.
>
> **Additional Ablation Studies** We investigate the effect of $\lambda_{code}$ in Table 2 of the supplementary. The attached pdf also depicts the t-SNE visualization of CIFAR10 instances for different features generated by our model. As can be seen in the figures, while our model’s features form separate clusters, our label embedding makes these clusters distinctive from each other.  Binary embedding enhances this separation while condensing the cluster by making samples of clusters closer to each other, which is evident for the yellow or bird cluster.  This is due to the fact that binary embedding, because of its 0 or 1 nature, will be affected by semantic similarity more than visual similarity. Finally, our code embedding shows indirectly that to have the most efficient code, our model should span the code space as much as possible, which explains the porous nature of these clusters. We will add the effects of different hyperparameters on these clusters to our paper.
>
> **Comparison with Sota** We made a few minor changes to the implementation, including using tanh instead of sigmoid activation functions for a more symmetric 0 and 1 code distribution, using a bias of 1 for masker to make the masker network start from considering all code bits at the beginning of training and finally using a label smoothing hyperparameter for preventing the network from being too strict in its code predictions. Our results based on these changes are in the table below with a comparison to recent Sota methods. We compare against recent GCD methods in the following table; we leave comparisons with methods with a different problem as NCD like [2,4] or different data like intent classification in [3] for future work.
> |**Dataset** |||CUB|||Aircraft|||Herb|||SCars|||Pet||
> |----------------------|----------------------|----------|----------|----------|----------|----------|----------|----------|----------|----------|----------|----------|----------|----------|----------|----------|
> |**Method** |Venue|All | Known  | Novel |  All | Known  | Novel|All | Known  | Novel|All | Known  | Novel|All | Known  | Novel|
> |k-means|ACM'07|34.3|38.9|32.1|12.9|12.9|12.8|13.0|12.2|13.4 |12.8|10.6|13.8|77.1|70.1|80.7|
> |RankStats+|TPAMI'21|33.3|51.6|24.2|26.9| 36.4| 22.2|27.9|55.8|12.8 |28.3|61.8|12.1|-|-|-|
> |UNO+|ICCV'21|35.1|49.0|28.1|40.3| 56.4| 32.2|28.3|53.7|14.7|35.5|70.5|18.6|-|-|	-|
> |ORCA|ICLR'22|36.3|43.8|32.6|31.6|32.0|31.4|24.6|26.5|23.7|31.9|42.2|26.9|	-|-|- |
> |GCD|CVPR'22|51.3|56.6|48.7|45.0	|41.1|46.9|35.4|51.0|27.0|39.0|57.6|29.9|	80.2	|85.1|77.6|
> |XCon [a]|BMVC'22|52.1|54.3|51.0|47.7|44.4|49.4|-|-|-|40.5|58.8|31.7|86.7 |91.5|84.1
> |PromptCAL[b]|CVPR'23|62.9|64.4|62.1|52.2|52.2|52.3|-|-|-|50.2|70.1|40.6|-|-|-|
> |DCCL [c] |CVPR'23|63.5|60.8|**64.9**|-|-|-|-|-|-|43.1|55.7|36.2|88.1|88.2|88.0|
> |SimGCD [d]|ICCV'23|60.3|65.6|57.7|54.2|59.1|51.8|**44.0**| 58.0|**36.4**|53.8|**71.9**|45.0|-|-|-|
> |GPC [e] |ICCV'23|52.0|55.5|47.5| 43.3|40.7|44.8|-|-|-| 38.2| 58.9|27.4|-|-|-|
> |InfoSieve |  | **70.9** | **83.5** |64.3|**60.6**| **69.1**| **56.4**| 40.3 |**59.0**| 30.2|**63.6**| 61.0 |**64.9**|**90.7**|**95.2**|**88.4**|
>
> [a] Fei, Yixin, et al. "Xcon: Learning with experts for fine-grained category discovery." 33rd British Machine Vision Conference. BMVA Press 2022.
>
> [b] Zhang, Sheng, et al. "Promptcal: Contrastive affinity learning via auxiliary prompts for generalized novel category discovery." Proceedings of the IEEE/CVF Conference on Computer Vision and Pattern Recognition. 2023.
>
> [c] Pu, Nan, Zhun Zhong, and Nicu Sebe. "Dynamic Conceptional Contrastive Learning for Generalized Category Discovery." Proceedings of the IEEE/CVF Conference on Computer Vision and Pattern Recognition. 2023.
>
> [d] Wen, Xin, Bingchen Zhao, and Xiaojuan Qi. "A Simple Parametric Classification Baseline for Generalized Category Discovery." Proceedings of the IEEE/CVF International Conference on Computer Vision. (2022).
>
> [e] Zhao, Bingchen, Xin Wen, and Kai Han. "Learning Semi-supervised Gaussian Mixture Models for Generalized Category Discovery." Proceedings of the IEEE/CVF International Conference on Computer Vision. (2023).
>
> **Transductive or instructive.** We use the same experimental setup as GCD and DCCL, which is transductive. We have a disjoint validation set to determine the best hyperparameters and best model.
>
> **Mutual information computation.** As mentioned in the supplemental sections 1.2.2 and 1.2.3, we can consider the reconstruction loss or contrastive loss to approximately maximize mutual information. We use contrastive loss since we base our approach on the GCD framework. Using MINE to optimize this mutual information aligns well with our theory, but it is computationally expensive in practice. However, estimating mutual information of code and categories can be feasible.

---

> > ### Comment · Reviewer_GHXK · 2023-08-16
> > **Reply to rebuttal**
> >
> > I read the paper again and checked all reviewers' comments and the author's reply. And I keep my score.

---

> > > ### Author Response · Authors · 2023-08-16
> > >
> > > We sincerely appreciate the time and effort you invested in your comprehensive review and your valuable suggestions, especially regarding the additional ablation studies and comparisons with recent state-of-the-art works. We have diligently revised our paper to incorporate these new experimental results. Should there be any further points of enhancement that could potentially lead to a positive reevaluation of our paper, we would greatly appreciate your guidance. Your constructive feedback is instrumental in elevating the quality of our work, and we are truly thankful for it.

---

> > > > ### Comment · Reviewer_GHXK · 2023-08-18
> > > > **Reply to author**
> > > >
> > > > Thank you for your response. However, the issues of over-parameterization and incomplete ablation studies still leave me unconvinced.
> > > > And I'm still not clear enough on how you perform binary classification. I also may have misunderstood on Theorem 1. Could you provide me with a more detailed explanation? I am happy to discuss with you.

---

> > > > > ### Author Response · Authors · 2023-08-21
> > > > >
> > > > > ***We thank the reviewer for engagement and offering us the opportunity for further clarification.***
> > > > >
> > > > > **Parameter ablation:** We agree that our model has a few hyperparameters, but let’s expand on these parameters: code binary constraint, mask binary constraint, code contrastive, code length, code mapping. We examine the effect of each hyperparameter on overall performance of the model on CIFAR10 and CUB datasets. Due to time constraints, we report the results for 20 epochs on CUB and 100 epochs on CIFAR10. Our default values for the hyperparameters are: code constraint coeff$=10$, mask constraint coeff$=10$, code contrastive coeff $\beta=1$, code mapping coeff $\gamma=1$ and code length coeff $\delta=0.1$.
> > > > >
> > > > > **Code binary constraint:** This hyperparameter is introduced to satisfy the binary requirement of the code. By using binary neural networks and an STE (straight-through estimator) [i], we can simply omit the requirement for this hyperparameter. Another approach would be to benefit from Boltzmann machines [ii] to have a binary code. The following table reports the results for different hyperparameters for this binary code condition. Note that since we use tanh to create the binary vector, the coefficient only determines how fast the method makes the conditions met, and when 0 and 1 are stabilized, the hyperparameter effect will be diminished. However, we noticed that bigger coefficients affect the known accuracy to some degree.
> > > > > |**Dataset**||CUB||
> > > > > |----------------------|----------|----------|----------|
> > > > > |Code constraint coeff|All |Known|Novel|
> > > > > |1|67.9|82.4|60.7|
> > > > > |10|68.0|82.3|60.9|
> > > > > |100|68.3|81.4|61.8|
> > > > > |1000|67.3|80.2|60.9|
> > > > >
> > > > > **Mask binary constraint:** For the mask constraint hyperparameter, again we can use a binary equivalent like binary neural network or Boltzmann machine. In our Lagrange multiplier approach, we start from an all-one mask, hence a bigger constraint translates to a bigger code length for categories. Since smaller or longer than required category lengths are both detrimental to the performance, a middle coefficient is more useful.
> > > > >
> > > > > |**Dataset**||CUB||
> > > > > |----------------------|----------|----------|----------|
> > > > > |Mask constraint coeff|All|Known|Novel|
> > > > > |1|69.9|81.9|63.9|
> > > > > |10|68.0|82.3|60.9|
> > > > > |100|71.3|80.3|66.8|
> > > > > |1000|68.6|81.6|62.1|
> > > > >
> > > > > **Code contrastive:** This loss is the loss that maintains information about the input. From the following table we observe that for a fine-grained dataset like CUB, minimizing the information will lead to a better performance, however for a generic dataset like CIFAR10, maintaining some information about the input itself can be beneficial especially for novel categories.
> > > > >
> > > > > |**Dataset**||CUB|||CIFAR|10|
> > > > > |----------------------|----------|----------|----------|----------|----------|----------|
> > > > > |$\beta$|All|Known|Novel|All |Known |Novel|
> > > > > |0.01|70.2|83.6|63.5|94.1|96.8|92.7|
> > > > > |0.1|66.4|79.8|59.7|94.7|96.8|93.6|
> > > > > |1|68.0|82.3|60.9|95.8|96.8|95.2|
> > > > > |2|67.7|80.5|61.3|95.4|96.8|94.8|
> > > > >
> > > > > **Code length:** Here we report our results for different code length hyperparameters, note that since the code length is penalized exponentially, the code length effect is not comparable to the exponential growth. However from this table it can be seen that the shorter code lengths (higher code length penalty), translates to a better novel category identification. Also note that since CIFAR 10 is a generic dataset, some extra information can actually help the model to distinguish categories from each other, while for the CUB dataset, it is better to omit the extra information.
> > > > >
> > > > > |**Dataset**||CUB|||CIFAR|10|
> > > > > |----------------------|----------|----------|----------|----------|----------|----------|
> > > > > |$\delta$|All| Known|Novel|All|Known|Novel|
> > > > > |0.01|66.5|79.2|60.2|94.8|96.7|93.8|
> > > > > |0.1|68.0|82.3|60.9|95.8|96.8|95.2|
> > > > > |1|70.1|82.1|64.1|94.8|96.8|93.9|
> > > > > |2|69.1|80.6|63.4|95.3|96.8|94.6|
> > > > >
> > > > > **Code mapping:** Since current evaluation metrics rely on hit-and-miss accuracy, our category codes must be mapped to this scenario. This loss is not an essential part of the self-coding that our model learns, but it accelerates the model’s training.
> > > > >
> > > > > |**Dataset**||CUB|||CIFAR|10|
> > > > > |----------------------|----------|----------|----------|----------|----------|----------|
> > > > > |$\gamma$|All|Known|Novel|All|Known| Novel|
> > > > > |0.01|69.3|82.1|62.9|95.0|96.8|94.1|
> > > > > |0.1|68.3|80.2 |62.4|94.9|96.6 |94.0|
> > > > > |1|68.0| 82.3|60.9|95.8|96.8|95.2|
> > > > > |2|68.6| 80.3 |62.7|95.4|96.8|94.6|
> > > > >
> > > > > One reason the model is not very sensitive to different hyperparameters is that our model consists of three separate parts: Code masker, Code Generator and Categorizer. The only hyperparameters that affect all of these three parts directly are $\beta$, the code contrastive coefficient, and $\delta$, the code length coefficient. This is why these hyperparameters have a much more pronounced effect on our model’s performance. We will include the ablation and discussion on the effect of the hyperparameters on our model in the updated manuscript.

---

> > > > > > ### Author Response · Authors · 2023-08-21
> > > > > >
> > > > > > [i] Bengio, Yoshua, Nicholas Léonard, and Aaron Courville. "Estimating or propagating gradients through stochastic neurons for conditional computation." arXiv preprint arXiv:1308.3432 (2013).
> > > > > >
> > > > > > [ii] Hinton, Geoffrey E. "A practical guide to training restricted Boltzmann machines." Neural Networks: Tricks of the Trade: Second Edition. Berlin, Heidelberg: Springer Berlin Heidelberg, 2012. 599-619.

---

> > > > > > > ### Author Response · Authors · 2023-08-21
> > > > > > >
> > > > > > > **Theorem 1 clarification:**
> > > > > > > The first part of theorem 1 states that if there is an implicit hierarchy tree, then for any category tree which is isomorph to this implicit tree, the algorithmic mutual information between each sample and its binary code generated by the tree will be maximum for the optimal tree. Hence maximizing this mutual information is a necessary condition for finding the optimal tree. This is equivalent to finding a tree that generates the shortest-length binary code to address each sample uniquely.
> > > > > > >
> > > > > > > The second part of theorem 1 states that for the optimal tree, the algorithmic mutual information between each sample category and its binary code will be maximum. Hence again, maximizing this mutual information is a necessary condition for finding the optimal tree. This is equivalent to finding a tree that generates the shortest-length binary code to address each category uniquely. This means that since the tree should be a valid tree, the prefix to the unique address of every sample of category $c$ should be the shortest-length binary code, while this shared prefix is not the prefix of any sample from other categories. We will update the manuscript to make this clearer.
> > > > > > >
> > > > > > > **Binary classification:**
> > > > > > > Suppose that the generated feature vector by the network for sample $X$ is $x_0x_1\cdots x_k$, where $k$ is the dimension of the network or equivalently the depth of our implicit hierarchy tree. Using appropriate activation functions, we can assume that $x_i$s are binary. The unsupervised contrastive loss forces the model to make the associated code to each sample unique. So if $X'$ is not equivalent to $X$ or one of its augmentations, its code $x'_0x'_1\cdots x'_k$ will differ from the code assigned to $X$. For the supervised contrastive loss, instead of considering the code, we consider a sequence by assigning different positional values to each bit so the code $x_0x_1\cdots x_k$ can be considered as binary number $(0.x_0x_1\cdots x_k)_2$. Then the supervised contrastive loss aims to minimize the difference between these assigned binary numbers. This means our model learns to use the first digits for discriminative information while pushing the specific information about each sample to the last digits. Then our masker learns to minimize the number of discriminative digits. Our theorem states that, finally, the embedded tree that the model learns this way is a good approximation of the optimal tree. In the end, our model generates a code for each sample, and we consider each code as a binary tree traverse from the root to the leaf, hence the codes delineate our tree's structure and binary classification that happens at each node. The effectivity of this tree structure extraction is evident on a dataset like CUB, where the discriminative aspects of images can be more nuanced than the augmentation of the images. Since our approach enables the model to use the initial bits for supervised contrastive learning and the last bits for unsupervised contrastive learning, we can benefit from their synergic advantages while preventing them from interfering with each other. We will further clarify the binary classification in the updated manuscript.

---

> > > > > > > > ### Comment · Reviewer_GHXK · 2023-08-22
> > > > > > > > **Reply to author**
> > > > > > > >
> > > > > > > > Thank you for your response. I'm pleased to see that you have addressed my concerns, and I'm inclined to vote for the acceptance of your manuscript.

---

> > > > > > > > > ### Author Response · Authors · 2023-08-22
> > > > > > > > >
> > > > > > > > > We are grateful for your insightful feedback. We believe that the added ablation studies and hyperparameter analyses have indeed enhanced our work, and we owe this improvement to your thorough and constructive suggestions. Thank you.

---

### Official Review · Reviewer_Sjrc · 2023-07-06

**Soundness:** 3 good
**Presentation:** 3 good
**Contribution:** 3 good
**Rating:** 7
**Confidence:** 5

**Summary:**

The paper conceptualize a category through the lens of optimization, viewing it as an optimal solution to a well-defined problem. The author propose a novel, efficient and self-supervised method capable of discovering previously unknown categories at test time.

**Strengths:**

1、The paper aims to tackle an important problem of how to better represent categories in deep learning models. The mathematical solution and novel framework proposed by the authors offer intriguing insights into addressing this issue;
2、The paper is well-structured, with a reasonable experimental design and demonstrates the effectiveness of the proposed method on multiple datasets.


**Weaknesses:**

1、It is worth noting that the proposed method might require more empirical studies to validate its generalizability across different scenarios and tasks. How does this model perform on the common ImageNet-100 dataset?
2、I agree that there are quite a few loss functions presented in this paper. Although the authors provide detailed explanations for each loss, it might be somewhat challenging to unify so many losses within a single system. So adapting this model for multi-stage training could be a viable approach to address the complexity if handling multiple losses.
3、The improvement on the commonly used CIFAR dataset is relative weak.


**Questions:**

1、In the Table. 3 and Table. 4, the performance gap between the method and sota methods to be quite significant. Can you analyze the possible reasons for this?
2、I am very interested in your implict binary tree and category encoding. Could you show it using a visualization?


**Limitations:**

 Authors adequately addressed the limitations .

---

> ### Author Rebuttal · Authors · 2023-08-09
>
> ***We thank the reviewer for finding the category definition an important problem, our framework novel, and our experimental design effective. We also agree that a theoretical foundation requires adequate empirical studies. Hence here we address the points raised by the reviewer to enrich our paper with more in-depth analysis and better visualizations.***
>
> **Limited improvement on CIFAR 10 and experiments on ImageNet100.** This is due to the limited number of categories in CIFAR 10. Our method is well suited for datasets with more categories and fine distinctions.
> Based on our theory, the improvement in CIFAR10/100 is predictable. For CIFAR 10, the depth of the implicit tree is 4, and for CIFAR 100, it will be 7; hence the number of implicit possible binary trees with this limited depth is smaller, meaning finding a good approximation for the implicit category tree can be achieved by other models. However, as the depth of this tree increases, our model can still find the aforementioned tree. This, in particular, is suitable for real-world scenarios where the number of categories can be huge, or categories are long-tailed like Herbarium_19, fine-grained like CUB, Aircraft, and Stanford Cars, partially labeled or have noisy labels. Nevertheless, we made a few minor changes to the implementation of our method and reported the new results on both generic and fine-grained datasets. Namely, we use tanh instead of the sigmoid activation functions for a more symmetric 0 and 1 code distribution, start with a bias of 1 for the masker to make the masker network start from considering all code bits at the beginning of training, and use a label smoothing hyperparameter for preventing the network from being too strict in its code predictions. Based on these changes, we see in the following table that our method is well-suited to fine-grained datasets.
> |**Dataset** |||CUB|||Aircraft|||Herb|||SCars|||Pet||
> |----------------------|----------------------|----------|----------|----------|----------|----------|----------|----------|----------|----------|----------|----------|----------|----------|----------|----------|
> |**Method** |Venue|All | Known  | Novel |  All | Known  | Novel|All | Known  | Novel|All | Known  | Novel|All | Known  | Novel|
> |k-means|ACM'07|34.3|38.9|32.1|12.9|12.9|12.8|13.0|12.2|13.4 |12.8|10.6|13.8|77.1|70.1|80.7|
> |RankStats+|TPAMI'21|33.3|51.6|24.2|26.9| 36.4| 22.2|27.9|55.8|12.8 |28.3|61.8|12.1|-|-|-|
> |UNO+|ICCV'21|35.1|49.0|28.1|40.3| 56.4| 32.2|28.3|53.7|14.7|35.5|70.5|18.6|-|-|	-|
> |ORCA|ICLR'22|36.3|43.8|32.6|31.6|32.0|31.4|24.6|26.5|23.7|31.9|42.2|26.9|	-|-|- |
> |GCD|CVPR'22|51.3|56.6|48.7|45.0	|41.1|46.9|35.4|51.0|27.0|39.0|57.6|29.9|	80.2	|85.1|77.6|
> |XCon [a]|BMVC'22|52.1|54.3|51.0|47.7|44.4|49.4|-|-|-|40.5|58.8|31.7|86.7 |91.5|84.1
> |PromptCAL[b]|CVPR'23|62.9|64.4|62.1|52.2|52.2|52.3|-|-|-|50.2|70.1|40.6|-|-|-|
> DCCL [c] |CVPR'23|63.5|60.8|**64.9**|-|-|-|-|-|-|43.1|55.7|36.2|88.1|88.2|88.0|
> |SimGCD [d]|ICCV'23|60.3|65.6|57.7|54.2|59.1|51.8|**44.0**| 58.0|**36.4**|53.8|**71.9**|45.0|-|-|-|
> |GPC [e] |ICCV'23|52.0|55.5|47.5| 43.3|40.7|44.8|-|-|-| 38.2| 58.9|27.4|-|-|-|
> |InfoSieve |  | **70.9** | **83.5** |64.3|**60.6**| **69.1**| **56.4**| 40.3 |**59.0**| 30.2|**63.6**| 61.0 |**64.9**|**90.7**|**95.2**|**88.4**|
>
> The new results on CIFAR10 and CIFAR100 (also ImageNet for 50 instead of 200 epochs due to the time limit) are as follows and will be added to the main paper.
> |**Dataset**|||CIFAR|10||CIFAR|100||Imagenet|100|
> |----------------------|----------------------|----------|----------|----------|----------|----------|----------|----------|----------|----------|
> |**Method**|Venue  | All | Known  | Novel |  All | Known  | Novel|All | Known  | Novel|
> |k-means |ACM'07| 83.6 | 85.7 | 82.5 | 52.0 | 52.2  | 50.8|72.7|75.5|71.3|
> |RankStats+|TPAMI'21 | 46.8 | 19.2 | 60.5| 58.2 | 77.6 | 19.3|37.1|61.6|24.8|
> |UNO+ |ICCV'21| 68.6 | **98.3** | 53.8 | 69.5 | 80.6  | 47.2|70.3|**95.0**	|57.9 |
> |ORCA |ICLR'22| 96.9 | 95.1 | 97.8 | 74.2 | 82.1  | 67.2|79.2	|93.2|72.1|
> |GCD|CVPR'22   | 91.5 | 97.9 | 88.2 | 73.0 | 76.2  | 66.5|74.1 | 89.8|66.3|
> |XCon [a]|BMVC'22 |96.0|97.3|95.4|74.2|81.2|60.3|77.6| 93.5 |69.7|
> |PromptCAL [b] |CVPR'23| **97.9** | 96.6 | **98.5**|**81.2**| **84.2**| 75.3|**83.1**|92.7|**78.3** |
> |DCCL [c]|CVPR'23| 96.3 | 96.5 | 96.9 | 75.3 | 76.8  | 70.2|80.5|90.5|76.2|
> |SimGCD [d] |ICCV'23 |97.1 | 95.1 | 98.1 | 80.1 | 81.2 | **77.8**|83.0|93.1|	77.9|
> |GPC [e] |ICCV'23|90.6| 97.6| 87.0| 75.4 |84.6| 60.1|75.3| 93.4 | 66.7|
> |InfoSieve| | 96.8 |96.4 |96.9| 77.8 | 80.5 | 72.4|78.8 | 90.3 |73.1|
>
> **Large performance gap for fine-grained datasets** We acknowledge the gap with the GCD paper. However, we'd like to emphasize that this gap has been closed by recent advances in state-of-the-art (SotA) methods, which we demonstrate in the table below. Like these new SotA methods, our method uses a richer set of supervision signals and tunable parameters. This approach accelerates the formation of supervised clusters and significantly increases the gap between the GCD paper over known categories.
>
> **Demonstration of binary codes** To demonstrate how the binary codes work, we provide a visualization in the attached pdf. Specifically, we include several sections of the binary tree our method obtains for the CUB dataset. We see that our method is able to extract some hierarchical structure in the data. As we can see in the leaves of this tree, we have “yellow sparrows” while they are the descendants of a more general “sparrow” node. Or for instance the path ‘00...1’ we have “black birds” node which can encompass multiple species of birds that are black.

---

> > ### Comment · Reviewer_Sjrc · 2023-08-15
> > **Response**
> >
> > Thank you for providing a rebuttal response, it has majorly addressed my concerns. The demonstration is excellent and  I will keep my current score.

---

> > > ### Author Response · Authors · 2023-08-16
> > >
> > > We are pleased to hear that our responses have addressed your concerns. We sincerely thank the reviewer for the thorough examination and invaluable suggestions that have contributed to the enhancement of our paper. We are committed to revising the paper to incorporate the new experimental results. We would greatly appreciate any additional feedback to further refine and enrich our paper.
> > > Thank you once again for your insightful comments.

---

### Official Review · Reviewer_ApGA · 2023-07-09

**Soundness:** 3 good
**Presentation:** 3 good
**Contribution:** 3 good
**Rating:** 6
**Confidence:** 3

**Summary:**

This paper questions a foundation problem that underpins the generalized class discovery (GCD): what defines a category. To this end, the authors proposed a self-learning-based method that solves GCD by modeling the implicit category hierarchy in the data. The proposed method allows control over category granularity and is thus good at fine-grained recognition datasets. The extensive experiments and theoretical analysis have demonstrated the effectiveness of the proposed method.

**Strengths:**

- The writing is satisfying. The paper is easy-to-follow.
- The motivation is decent.
- The proposed method is novel and proved effective.
- Also the related works are adequately discussed.

**Weaknesses:**

- The compared methods are no longer state-of-the-art. Also, due to the explicit assumption of a hierarchical structure, the proposed method is not always the best on all datasets, especially the generic ones.

**Questions:**

No questions.

--- Post rebuttal ---

Thank the authors for the detailed rebuttal response, which addressed my main concerns, and thus I keep my positive rating unchanged.

**Limitations:**

The limitations are carefully discussed in the paper.

---

> ### Author Rebuttal · Authors · 2023-08-09
>
> ***We thank the reviewer for finding our proposed method novel and effective and our motivation decent. We also agree that a strong theoretical approach will be more applicable if it is accompanied by strong empirical results. Hence here, we provide the empirical results that the reviewer has mentioned.***
>
> **Recent state-of-the-art methods.** Here, we add a comparison to several recent works, Xcon [a], PromptCAL [b], DCCL[c], SimGCD [d], and GPC [e], over fine-grained data. By making a few minor changes to the implementation of our method, we outperform all of these works on CUB-200, FGCV-Aircraft, Herbarium-19, Stanford-Cars, and Oxford-IIIT Pet. Namely, we use tanh instead of the sigmoid activation functions for a more symmetric 0 and 1 code distribution, use a bias of 1 for the masker to make the masker network start from considering all code bits at the beginning of training, alternate the training of coder and masker to separate their effect as suggested by reviewer Sjrc and use a label smoothing hyperparameter for preventing the network from being too strict in its code predictions. Based on these changes, our results are as follows and will be added to the main paper.
> |**Dataset** |||CUB|||Aircraft|||Herb|||SCars|||Pet||
> |----------------------|----------------------|----------|----------|----------|----------|----------|----------|----------|----------|----------|----------|----------|----------|----------|----------|----------|
> |**Method** |Venue|All | Known  | Novel |  All | Known  | Novel|All | Known  | Novel|All | Known  | Novel|All | Known  | Novel|
> |k-means|ACM'07|34.3|38.9|32.1|12.9|12.9|12.8|13.0|12.2|13.4 |12.8|10.6|13.8|77.1|70.1|80.7|
> |RankStats+|TPAMI'21|33.3|51.6|24.2|26.9| 36.4| 22.2|27.9|55.8|12.8 |28.3|61.8|12.1|-|-|-|
> |UNO+|ICCV'21|35.1|49.0|28.1|40.3| 56.4| 32.2|28.3|53.7|14.7|35.5|70.5|18.6|-|-|	-|
> |ORCA|ICLR'22|36.3|43.8|32.6|31.6|32.0|31.4|24.6|26.5|23.7|31.9|42.2|26.9|	-|-|- |
> |GCD|CVPR'22|51.3|56.6|48.7|45.0	|41.1|46.9|35.4|51.0|27.0|39.0|57.6|29.9|	80.2	|85.1|77.6|
> |XCon [a]|BMVC'22|52.1|54.3|51.0|47.7|44.4|49.4|-|-|-|40.5|58.8|31.7|86.7 |91.5|84.1
> |PromptCAL[b]|CVPR'23|62.9|64.4|62.1|52.2|52.2|52.3|-|-|-|50.2|70.1|40.6|-|-|-|
> DCCL [c] |CVPR'23|63.5|60.8|**64.9**|-|-|-|-|-|-|43.1|55.7|36.2|88.1|88.2|88.0|
> |SimGCD [d]|ICCV'23|60.3|65.6|57.7|54.2|59.1|51.8|**44.0**| 58.0|**36.4**|53.8|**71.9**|45.0|-|-|-|
> |GPC [e] |ICCV'23|52.0|55.5|47.5| 43.3|40.7|44.8|-|-|-| 38.2| 58.9|27.4|-|-|-|
> |InfoSieve |  | **70.9** | **83.5** |64.3|**60.6**| **69.1**| **56.4**| 40.3 |**59.0**| 30.2|**63.6**| 61.0 |**64.9**|**90.7**|**95.2**|**88.4**|
>
> Based on our theory, the improvement in CIFAR10/100 and ImageNet-100 is predictable. For CIFAR 10, the depth of the implicit tree is 4, and for CIFAR 100, it will be 7; hence the number of implicit possible binary trees with this limited depth is smaller, meaning finding a good approximation for the implicit category tree can be achieved by other models. However, as the depth of this tree increases, our model can still find the aforementioned tree. This, in particular, is suitable for real-world scenarios where the number of categories can be huge, or categories are long-tailed like Herbarium_19, fine-grained like CUB, Aircraft, and Stanford Cars, partially labeled or have noisy labels. Nevertheless, we repeated experiments with the aforementioned changes and reported the new results on CIFAR10 and CIFAR100 (also ImageNet for 50 out 200 epochs due to the time limit). Based on these changes, our results are as follows and will be added to the main paper.
> |**Dataset**|||CIFAR|10||CIFAR|100||Imagenet|100|
> |----------------------|----------------------|----------|----------|----------|----------|----------|----------|----------|----------|----------|
> |**Method**|Venue  | All | Known  | Novel |  All | Known  | Novel|All | Known  | Novel|
> |k-means |ACM'07| 83.6 | 85.7 | 82.5 | 52.0 | 52.2  | 50.8|72.7|75.5|71.3|
> |RankStats+|TPAMI'21| 46.8 | 19.2 | 60.5| 58.2|77.6 | 19.3|37.1|61.6|24.8|
> |UNO+ |ICCV'21| 68.6 | **98.3** | 53.8 | 69.5 | 80.6  | 47.2|70.3|**95.0**	|57.9 |
> |ORCA |ICLR'22| 96.9 | 95.1| 97.8 |74.2 |82.1| 67.2|79.2	|93.2|72.1|
> |GCD|CVPR'22| 91.5 | 97.9 | 88.2 |73.0 |76.2| 66.5|74.1 | 89.8|66.3|
> |XCon [a]|BMVC'22 |96.0|97.3|95.4|74.2|81.2|60.3|77.6| 93.5 |69.7|
> |PromptCAL [b] |CVPR'23| **97.9** | 96.6 | **98.5**|**81.2**|**84.2**| 75.3|**83.1**|92.7|**78.3** |
> |DCCL [c]|CVPR'23| 96.3 | 96.5 | 96.9 | 75.3 | 76.8  | 70.2|80.5|90.5|76.2|
> |SimGCD [d] |ICCV'23 |97.1 | 95.1 | 98.1 | 80.1 | 81.2 | **77.8**|83.0|93.1|77.9|
> |GPC [e] |ICCV'23|90.6| 97.6| 87.0| 75.4 |84.6| 60.1|75.3| 93.4 | 66.7|
> |InfoSieve| | 96.8 |96.4 |96.9| 77.8 | 80.5 | 72.4|78.8 | 90.3 |73.1|
>
> Although our method is not always the best, it performs consistently well. While our model on fine-grained datasets outperforms most recent SoTAs, our model's performance on the generic datasets has not been compromised.

---

> > ### Comment · Reviewer_ApGA · 2023-08-16
> >
> > Thanks the authors for the detailed experiemental results and analysis. My main concerns are mostly addressed. I keep my rating for now before discussion with other reviewers.

---

> > > ### Author Response · Authors · 2023-08-16
> > >
> > > We are pleased to learn that we have successfully addressed your concerns. We extend our gratitude for your insightful suggestions, particularly regarding comparisons with the recent state of the art. In our endeavor to continually elevate the quality of our research, we have revised the paper, incorporating new experimental results spanning 5 fine-grained datasets and 3 generic datasets. Should you identify any additional areas for refinement or have insights that could further enhance the quality of our work, your guidance would be immensely valuable. We are thankful for your positive assessment and we remain committed to benefiting from your constructive feedback to continue elevating our paper.

---

### Official Review · Reviewer_xjo1 · 2023-07-11

**Soundness:** 3 good
**Presentation:** 3 good
**Contribution:** 3 good
**Rating:** 5
**Confidence:** 5

**Summary:**

This paper tackles the problem of generalized category discovery (GCD) by reframing the concept of a category with an implicit category code tree, which addresses three problems of conventional supervised learning, namely category hierarchies, label inconsistency, and open-world recognition. The contributions of this paper are: (1) conceptualizing a category with a theoretically optimal solution; (2) proposing a practical GCD method based on the theory; (3) achieving the state-of-the-art GCD performance on both generic and fine-grained scenarios.

**Strengths:**

(1) The inspiration for modeling category hierarchies by encoding a category with a binary tree is reasonable.

(2) The theoretical derivation of the optimization loss is adequate and helps readers understand the subsequent model design.

**Weaknesses:**

(1) The experimental demonstration of how binary code works is missing. For example, the authors can simply compare the codes of “dog”, “cat”, and “car” to see how the number of digits differs in the codes. It is crucial to verify the category hierarchies and label consistency, which are the core problems to solve claimed in the paper.

(2) The performance improvement on generic datasets is limited. CIFAR10 performance lags behind ORCA and CIFAR100 performance gains only 0.2.

(3) The paper's ablation study in Table 2 raises concern. In the first row, the model solely uses the visual features to compute the loss, i.e., the original GCD [1] approach. In Table 3 CIFAR10 section, the results provided from the original GCD paper are used. If we compare the results from both Table 2 and Table 3 by evaluating them using Semi-supervised K-means (SS-KMeans [1]), both InfoSieve and GCD approaches gain an increase in accuracy for the known class, while for the novel class, only InfoSieve gains an increase, while GCD gets the worst accuracy results. This paper shows that the GCD + generic K-Means is better than GCD + SS-KMeans, which is inconsistent with what the GCD paper claimed. An investigation should be conducted to address this inconsistency.

Minor: Missing notation definitions such as "M" and "K" in the main paper.

**Questions:**

Please see the weaknesses above.

In addition, should the “Z^i” in line 215 be “z^i”? Because “Z^i” is a number according to line 150?

**Limitations:**

yes

---

> ### Author Rebuttal · Authors · 2023-08-09
>
> ***We thank the reviewer for finding our inspiration and theoretical derivation reasonable and adequate. We use the points raised by the reviewer for better clarification of our theory and implementation details. The binary tree visualization and discussing the fine-grained vs. generic dataset performance will be a valuable addition to our paper.***
>
> **Demonstration of binary codes.** The attached pdf shows how the binary codes work for the CUB dataset. We see that our method can extract some hierarchical structure in the data. In the leaves of this tree, we have “yellow sparrows,” which are the descendants of a more general “sparrow” node. Or, for instance, in the path ‘00...1’, we have “black birds” node, which can encompass multiple black bird species.
>
>  **Limited improvement on CIFAR 10.** This is due to the limited number of categories in CIFAR 10. Our method is well suited for datasets with more categories and fine distinctions.
> Based on our theory, the improvement in CIFAR10/100 is predictable. For CIFAR 10, the depth of the implicit tree is 4; hence the number of implicit possible binary trees with this limited depth is smaller, meaning finding a good approximation for the implicit category tree can be achieved by other models. However, as the depth of this tree increases, our model can still find the aforementioned tree. This, in particular, is suitable for fine-grained data. For instance, the following table shows our results on fine-grained datasets:
> |**Dataset** |||CUB|||Aircraft|||Herb|||SCars|||Pet||
> |----------------------|----------------------|----------|----------|----------|----------|----------|----------|----------|----------|----------|----------|----------|----------|----------|----------|----------|
> |**Method**|Venue|All|Known|Novel|All|Known|Novel|All |Known|Novel|All|Known|Novel|All|Known|Novel|
> |k-means|ACM'07|34.3|38.9|32.1|12.9|12.9|12.8|13.0|12.2|13.4 |12.8|10.6|13.8|77.1|70.1|80.7|
> |RankStats+|TPAMI'21|33.3|51.6|24.2|26.9| 36.4| 22.2|27.9|55.8|12.8 |28.3|61.8|12.1|-|-|-|
> |UNO+|ICCV'21|35.1|49.0|28.1|40.3|56.4| 32.2|28.3|53.7|14.7|35.5|70.5|18.6|-|-|-|
> |ORCA|ICLR'22|36.3|43.8|32.6|31.6|32.0|31.4|24.6|26.5|23.7|31.9|42.2|26.9|-|-|-|
> |GCD|CVPR'22|51.3|56.6|48.7|45.0|41.1|46.9|35.4|51.0|27.0|39.0|57.6|29.9|80.2|85.1|77.6|
> |XCon [a]|BMVC'22|52.1|54.3|51.0|47.7|44.4|49.4|-|-|-|40.5|58.8|31.7|86.7 |91.5|84.1
> |PromptCAL[b]|CVPR'23|62.9|64.4|62.1|52.2|52.2|52.3|-|-|-|50.2|70.1|40.6|-|-|-|
> DCCL [c]|CVPR'23|63.5|60.8|**64.9**|-|-|-|-|-|-|43.1|55.7|36.2|88.1|88.2|88.0|
> |SimGCD [d]|ICCV'23|60.3|65.6|57.7|54.2|59.1|51.8|**44.0**|58.0|**36.4**|53.8|**71.9**|45.0|-|-|-|
> |GPC [e]|ICCV'23|52.0|55.5|47.5| 43.3|40.7|44.8|-|-|-|38.2|58.9|27.4|-|-|-|
> |InfoSieve||**70.9**|**83.5**|64.3|**60.6**|**69.1**|**56.4**|40.3|**59.0**|30.2|**63.6**|61.0|**64.9**|**90.7**|**95.2**|**88.4**|
>
> Nevertheless, we made a few minor changes to the implementation of our method and reported the new results on CIFAR10 and CIFAR100 (also ImageNet for 50 epochs due to the time limit). Namely, we use tanh instead of the sigmoid activation functions for a more symmetric 0 and 1 code distribution, start with a bias of 1 for the masker to make the masker network start from considering all code bits at the beginning of training, and use a label smoothing hyperparameter for preventing the network from being too strict in its code predictions. Based on these changes, our results are as follows and will be added to the main paper.
> |**Dataset**|||CIFAR|10||CIFAR|100||Imagenet|100|
> |----------------------|----------------------|----------|----------|----------|----------|----------|----------|----------|----------|----------|
> |**Method**|Venue|All|Known|Novel|All|Known|Novel|All|Known|Novel|
> |k-means|ACM'07|83.6|85.7|82.5|52.0|52.2|50.8|72.7|75.5|71.3|
> |RankStats+|TPAMI'21|46.8|19.2|60.5|58.2|77.6|19.3|37.1|61.6|24.8|
> |UNO+|ICCV'21|68.6|**98.3**| 53.8|69.5|80.6|47.2|70.3|**95.0**|57.9|
> |ORCA|ICLR'22|96.9|95.1|97.8|74.2|82.1|67.2|79.2|93.2|72.1|
> |GCD|CVPR'22|91.5|97.9|88.2|73.0|76.2|66.5|74.1|89.8|66.3|
> |XCon [a]|BMVC'22|96.0|97.3|95.4|74.2|81.2|60.3|77.6|93.5|69.7|
> |PromptCAL [b]|CVPR'23|**97.9**|96.6|**98.5**|**81.2**|**84.2**|75.3|**83.1**|92.7|**78.3**|
> |DCCL [c]|CVPR'23|96.3|96.5|96.9|75.3|76.8|70.2|80.5|90.5|76.2|
> |SimGCD [d] |ICCV'23|97.1|95.1|98.1|80.1|81.2|**77.8**|83.0|93.1|77.9|
> |GPC [e] |ICCV'23|90.6|97.6|87.0|75.4|84.6|60.1|75.3|93.4|66.7|
> |InfoSieve||96.8|96.4|96.9|77.8|80.5|72.4|78.8|90.3|73.1|
>
> **GCD results.** The difference between the first row of Table 2 and the GCD results reported is the number of blocks finetuned in the ViT, GCD freezes the weights of ViT-B-16 for 11 blocks and finetunes only the final block. Like DCCL, we freeze the weights of 10 blocks and finetune the final two blocks, because our model requires more tunable parameters to optimize our objective function. When freezing 11 and finetuning the last block of ViT (similar to GCD) for our method, we obtain 95.4% for known categories and 91.0% for novel categories on CIFAR10 dataset. This is still an improvement over the 88.2% obtained by GCD for novel categories. In the paper, we will add the number of blocks fine-tuned to our implementation details.
>
> **Missing notation definitions.** $M$ is the length of the binary code assigned to a sample $X^i$. $K$ is the length of the path to a category code $c$. We will update lemma $1$ to explain these symbols explicitly.
>
> **$Z^i$ clarification.** We can consider $Z^i$, as the mapping of sequence $z^i$ with length $d$, so the $Z^i$ is a number in this line. We will make it explicit in the text.

---

### Author Rebuttal · Authors · 2023-08-09

We thank all reviewers for their insightful feedback, and we appreciate that the reviewers found our paper theoretically and methodologically novel. There were a few shared questions about our paper, both in terms of comparing against the recently published state-of-the-art and visualizing our method of implicit tree extraction. To this end, we add a comparison to several recent works, Xcon [a], PromptCAL [b], DCCL [c], SimGCD [d], and GPC [e]. Furthermore, the attached PDF shows a demonstration of how our model extracts an implicit category tree for samples of the CUB-200 dataset.

[a] Fei, Yixin, et al. "Xcon: Learning with experts for fine-grained category discovery." 33rd British Machine Vision Conference. BMVA Press 2022.

[b] Zhang, Sheng, et al. "Promptcal: Contrastive affinity learning via auxiliary prompts for generalized novel category discovery." Proceedings of the IEEE/CVF Conference on Computer Vision and Pattern Recognition. 2023.

[c] Pu, Nan, Zhun Zhong, and Nicu Sebe. "Dynamic Conceptional Contrastive Learning for Generalized Category Discovery." Proceedings of the IEEE/CVF Conference on Computer Vision and Pattern Recognition. 2023.

[d] Wen, Xin, Bingchen Zhao, and Xiaojuan Qi. "A Simple Parametric Classification Baseline for Generalized Category Discovery." Proceedings of the IEEE/CVF International Conference on Computer Vision. (2022).

[e] Zhao, Bingchen, Xin Wen, and Kai Han. "Learning Semi-supervised Gaussian Mixture Models for Generalized Category Discovery." Proceedings of the IEEE/CVF International Conference on Computer Vision. (2023).

---

### Decision · Program_Chairs · 2023-09-21

**Decision:**

Accept (poster)

**Comment:**

This paper tackles the generalized category discovery (GCD) problem, proposing an interesting perspective of learning an implicit category code tree. A key advantages of this is supporting category hierarchies and label noise, hence moving more towards open-world situations. The paper formulates this as an optimization problem and shows theoretical motivation for the solution, resulting in a practical algorithm (InfoSieve). Results are shown across a range of datasets, with especially strong performance on more fine-grained datasets with category hierarchies.

  The reviewers appreciated the interesting problem formulation, theoretical justification, and novelty of the method. They shared a number of concerns, however, including 1) an understanding of how binary codes actually show label consistency and hierarchies, 2) limited performance on smaller datasets with less rich categories (CIFAR-10), 3) more up-to-date comparison to state-of-art, and 4) the need for a large number of losses/hyper-parameters. The authors provided a strong rebuttal, including a number of different experiments on fine-grained datasets, improved results with a few slight modifications, and hyper-parameter sensitivity analysis. After the rebuttal, most of the reviewers were satisfied and kept or increased to positive scores.

  After reviewing the paper, reviews, and rebuttal, I recommend acceptance. One of the interesting and unique aspects of this work is its focus on a more structured, hierarchical understanding of categories in the novel category discovery setting, which will hopefully foster additional methods with this perspective.